# Through the Lens of Contrast: Self-Improving Visual Reasoning in VLMs

**Zhiyu Pan**[1,2*], **Yizheng Wu**[2*], **Jiashen Hua**[2†], **Junyi Feng**[2], **Shaotian Yan**[2], **Bing Deng**[2], **Zhiguo Cao**[1], **Jieping Ye**[2]

[1]Huazhong University of Science and Technology,  [2]Alibaba Cloud

## Abstract

Reasoning has emerged as a key capability of large language models. In linguistic tasks, this capability can be enhanced by self-improving techniques that refine reasoning paths for subsequent finetuning. However, extending these language-based self-improving approaches to vision language models (VLMs) presents a unique challenge: visual hallucinations in reasoning paths cannot be effectively verified or rectified. Our solution starts with a key observation about visual contrast: when presented with a contrastive VQA pair, *i.e.*, two visually similar images with synonymous questions, VLMs identify relevant visual cues more precisely. Motivated by this observation, we propose **V**isual **C**ontrastive **S**elf-**Ta**ught **R**easoner (VC-STaR), a novel self-improving framework that leverages visual contrast to mitigate hallucinations in model-generated rationales. We collect a diverse suite of VQA datasets, curate contrastive pairs according to multi-modal similarity, and generate rationales using VC-STaR. Consequently, we obtain a new visual reasoning dataset, VisCoR-55K, which is then used to boost the reasoning capability of various VLMs through supervised finetuning. Extensive experiments show that VC-STaR not only outperforms existing self-improving approaches but also surpasses models finetuned on the SoTA visual reasoning datasets, demonstrating that the inherent contrastive ability of VLMs can bootstrap their own visual reasoning. Project at: `https://github.com/zhiyupan42/VC-STaR`.

## 1 Introduction

The scaling of large language models (LLM) has led to the emergence of reasoning capabilities (Wei et al., 2022a), making a transition from *System* 1 to *System* 2 (Kahneman, 2011) and enabling language models to tackle complex, multi-step problems (Wei et al., 2022b; Kojima et al., 2022). This emergent ability can be further enhanced by various techniques (Wang et al., 2023b; Li et al., 2023b; Hao et al., 2023; Gao et al., 2023; OpenAI, 2024b; Guo et al., 2025). Among them, self-improving approaches (Zelikman et al., 2022; Gulcehre et al., 2023; Madaan et al., 2023; Qu et al., 2024; Ma et al., 2025) form a prominent branch, mainly because they can be easily applied and extended without external reward models (Lu et al., 2024a), predefined step decomposition (Liu et al., 2025), or specially designed reasoning structures (Li et al., 2025).

However, it is infeasible to directly adapt such language-based self-improving methods to vision language models (VLMs) (Liu et al., 2023; Bai et al., 2025). Previous self-improving approaches focus on textual coherence and the quality of the final answer (Zelikman et al., 2022; Zhang et al., 2024a), while they are unable to verify or rectify the visual hallucinations that persist in current VLMs (Tong et al., 2024; Li et al., 2024). Even worse, they may get stuck in speculative reasoning that privileges textual priors over real visual evidence (Favero et al., 2024; Wu et al., 2025). We claim that the key problem for self-improving in VLMs is: *how to rectify visual hallucinations in VLMs' reasoning paths for high-quality visual rationale generation.*

Our solution is built upon an interesting observation: *VLMs can see better during contrasting.* As shown in Fig. 1a, the VLM generates a wrong rationale with visual hallucinations given a single visual question answering (VQA) sample. Instead, when presented with a contrastive VQA pair,

---

*Same contribution.

†Corresponding author and project leader.

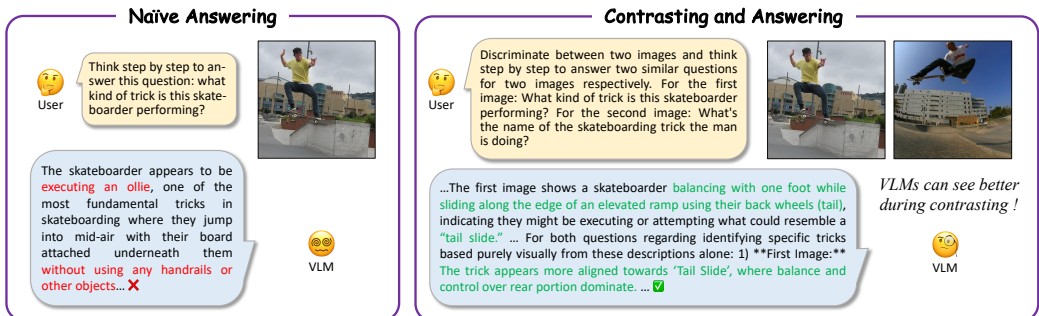

(a) Visual hallucinations within the reasoning paths can mislead the model. Contrasting within a contrastive VQA pair, the VLM may rectify its own hallucinations.

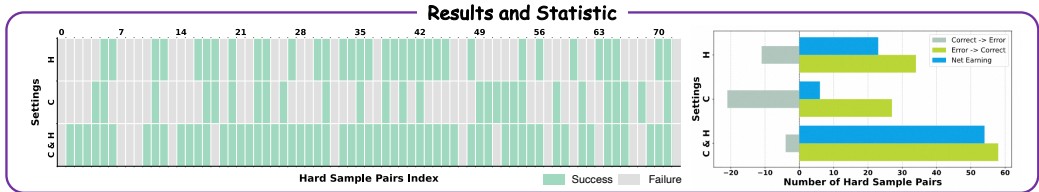

(b) Results and statistics of rectifying hallucinatory outputs by three settings. H: with hint; C: via contrasting.

Figure 1: **Contrasting makes the VLM see better.** (a) Contrastive VQA pairs compels a more accurate response. (b) Compared with a previous self-improving method STaR (Zelikman et al., 2022) that enhances the quality of reasoning with hints (ground-truth answers), contrasting with hints can rectify more cases. The blocks along the $x$-axis mark initial VLM failures. The color of each block indicates the outcome of rectifying: green for success and gray for failure. Tested VLM is Qwen2.5VL-7B (Bai et al., 2025).

*i.e.*, two similar images with synonymous questions (Setting C in sub-figure 1b), the model captures fine-grained visual evidence more accurately and rectifies the erroneous rationale. Statistics of this phenomenon on a group of failure cases are shown in Fig. 1b. Compared with the hints-only (Setting H in sub-figure 1b) self-improving (provide the model with the ground-truth answers), the hints and contrasting (Setting C&H in sub-figure 1b) setting not only prevents the model from making new errors but also leads to the rectification of its original hallucinations.

Motivated by this, we propose a new self-improving framework, **V**isual **C**ontrastive **S**elf-**Ta**ught **R**easoner (VC-STaR). VC-STaR contains three steps: (1) *think step by step* and generate a coarse rationale; (2) *compare* visual queries in a contrastive VQA pair and provide a contrastive analysis; (3) *rethink* and refine the coarse rationale via an LLM based on the contrastive analysis. In order to guarantee the scalability of VC-STaR, we also propose a task-agnostic contrastive VQA pair curation framework, which can be readily adapted to various VQA tasks, *e.g.*, reasoning (Lu et al., 2021b), math (Gao et al., 2025), chart Liu et al. (2024a), and OCR (Yuan et al., 2022). Specifically, we curate the contrastive VQA pairs within individual datasets, based on the similarity of both images and questions. We utilize these contrastive VQA pairs to generate faithful rationales, resulting in a novel **Vis**ual **Co**ntrastive **R**easoning dataset (VisCoR-55K) as illustrated in Fig. 2. Finetuning with VisCoR-55K enhances VLMs' visual reasoning capability.

VC-STaR achieves prominent results on a wide range of challenging benchmarks, including MMVP (Tong et al., 2024), HallusionBench (Guan et al., 2024), MathVista (Lu et al., 2024b), MathVision Wang et al. (2024), and MMStar (Chen et al., 2024c). On the one hand, VC-STaR outperforms existing self-improving baselines. On the other hand, it exhibits a clear advantage over models trained on recently proposed reasoning datasets. The experimental results validate that visual reasoning capability of VLMs can be bootstrapped through the lens of contrast.

## 2 RELATED WORKS

**Reasoning in Language.** Dual-system theory (Kahneman, 2011) illustrates two systems in human cognition: a fast, intuitive *System* 1 and a slow, deliberate *System* 2 which is akin to emergent

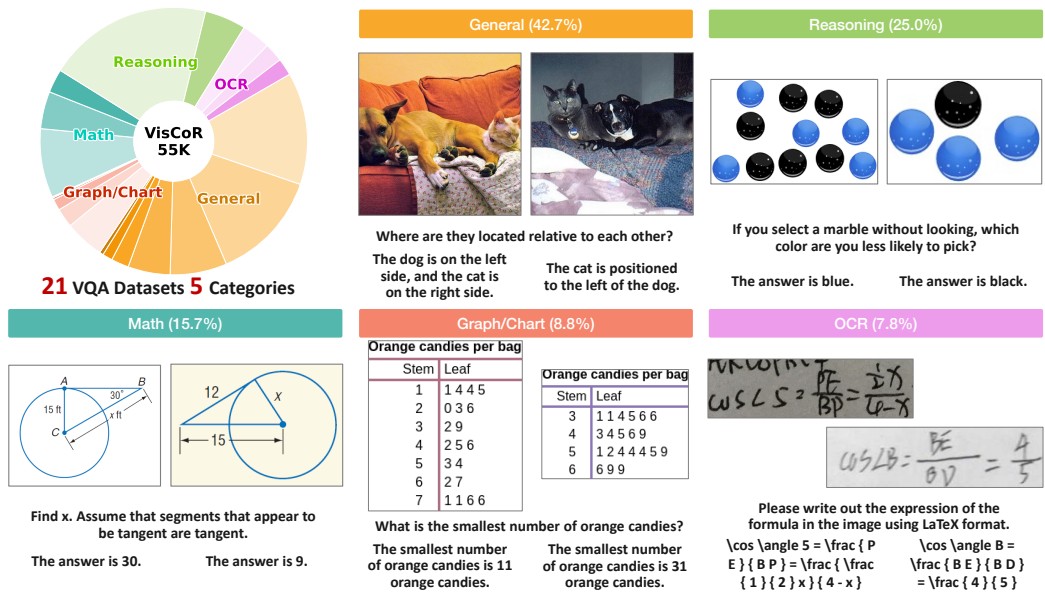

Figure 2: **VisCoR-55K.** We introduce the **Vis**ual **Co**ntrastive **R**easoning dataset (VisCoR-55K), a new collection of 55K high-quality visual reasoning samples. Spanning the domains of general VQA, reasoning, math, graph/chart, and OCR, each sample is created by leveraging a contrastive counterpart to generate a faithful rationale. Rationales are shown in the Sec. A.4.

reasoning capability of LLMs (Wei et al., 2022a). Consequently, reasoning enhancement (Wei et al., 2022b; Kojima et al., 2022) is considered a pathway to elevate LLMs' cognitive performance. One solution involves a reward model (Li et al., 2023b; Lu et al., 2024a), often coupled with Monte Carlo tree search (Hao et al., 2023; Zhang et al., 2024a), to discover optimal reasoning paths. However, this solution is constrained by the need of an auxiliary model and the requirement for reasoning step dividing (Liu et al., 2025). Another way employs macro reasoning actions (Gao et al., 2023; Khot et al., 2023; Yang et al., 2025a) to inject human prior knowledge, however, hand-crafted macro actions struggle to adapt to diverse reasoning scenarios. While reinforcement learning (Rafailov et al., 2023; Trung et al., 2024; Guo et al., 2025) has also attracted the attention, its success relies on the data format and the design of reward functions. Self-improving methods (Zhang et al., 2024a) offer a more scalable alternative, enabling LLMs to refine its own reasoning by constructing high-quality reasoning data (Wang et al., 2023b), utilizing ground-truth answers as hints (Zelikman et al., 2022), or leveraging internal feedback (Qu et al., 2024). With fewer external constraints, self-improving methods pave the way for more flexible and general language reasoners.

**Reasoning in Vision.** Human reasoning is stimulated not only by textual input but also by visually-related queries. Fostering the visual reasoning ability (Zhang et al., 2024c) for VLMs (Liu et al., 2023; Li et al., 2023a) is therefore a critical frontier topic. Early attempts often rely on external scaffolding like scene graphs (Mitra et al., 2024), macro actions (Xu et al., 2025; Dong et al., 2025), or bounding boxes highlighting key region in images (Shao et al., 2024). However, such approaches suffer from fundamental limitations: they are constrained by the data structure or tend to generate stereotyped reasoning paths. Despite these advances, the self-improving paradigm which has shown its effectiveness in text-only domain is underexplored for visual reasoning. The primary obstacle is the visual hallucinations embedded in reasoning paths cannot be easily rectified by existing text-centric self-improving frameworks (Zhang et al., 2024a; Zelikman et al., 2022; Qu et al., 2024). The proposed VC-STaR attempts to bridge this gap through the lens of contrast.

**Power of Contrasting.** Contrasting has shown effectiveness in a wide range of machine learning topics. By comparing different *views* (Tian et al., 2020), *e.g.*, data augmentations, of the same sample while distinguishing them from others (Wang & Isola, 2020), contrastive self-supervised learning methods (He et al., 2020; Grill et al., 2020; Radford et al., 2021; Liang et al., 2022; Pan et al., 2023) excel at learning potent feature representations. Explicitly cross-image contrasting is also studied under uni-modal setting (Pan et al., 2023; Ding et al., 2024; Chen et al., 2024a) and multi-modal setting (Park et al., 2019; Kim et al., 2021; Yao et al., 2022; Dunlap et al., 2024). Based

on these advancements, VLMs are endowed with robust capabilities for multi-image comprehension and comparison (Alayrac et al., 2022; Bai et al., 2025; Chameleon, 2025; Lin et al., 2025). Some prior works have leveraged contrasting to create better instruction-tuning data (Jiao et al., 2025; Ma et al., 2024). However, how contrasting can help visual reasoning remains an open question. We observe that VLMs' inherent comparative ability can be repurposed to actively suppress its own visual hallucinations, bootstrapping their visual reasoning capability. This discovery offers a new perspective about the power of contrasting in reasoning.

## 3 VISUAL CONTRASTIVE SELF-TAUGHT REASONER (VC-STAR)

Let $\theta$ be a VLM and $\mathcal{D} = \{(v_i, q_i, a_i)\}_{i=1}^N$ be a set of visual question answering (VQA). The VQA set consists $N$ triplets, where $v_i$, $q_i$, and $a_i$ represent the $i$-th image, question, and corresponding ground-truth answer, respectively. Following previous self-taught reasoners (Zelikman et al., 2022; Madaan et al., 2023), the original VQA dataset $\mathcal{D}$ can be enriched by generating a rationale $r$ with $\theta$ for each triplet, which transforms $\mathcal{D}$ into a visual reasoning dataset $\mathcal{R} = \{(v_i, q_i, a_i, r_i)\}_{i=1}^M$. However, as mentioned in Sec. 1, rationale $r_i$ may be contaminated by visual hallucinations. Motivated by the observation illustrated in Fig. 1, VC-STaR aims to refine rationale $r_i$ into a more faithful one $\tilde{r}_i$ by contrasting the $(v_i, q_i, a_i)$ with a contrastive VQA counterpart sample $(\hat{v}_i, \hat{q}_i, \hat{a}_i)$ where $q_i$ is synonymous with $\hat{q}_i$ and $v_i$ shares similar context with $\hat{v}_i$. The contrastive VQA pairs $\mathcal{P} = \{((v_i, q_i, a_i), (\hat{v}_i, \hat{q}_i, \hat{a}_i))\}_{i=1}^K$ support the contrasting and rationale refining process. The contrastive VQA pairs are curated by searching $(\hat{v}_i, \hat{q}_i, \hat{a}_i)$ for $(v_i, q_i, a_i)$ within diverse data groups in $\mathcal{D}$ for different VQA tasks, ensuring the generalization of VC-STaR. The VC-STaR is designed to address two key challenges: (1) how to curate meaningful contrastive VQA pairs; (2) how to transfer the fine-grained discriminative ability from dual-image contrasting to refine the single-image reasoning. Sec. 3.1 elaborates on the pipeline for curating contrastive VQA pairs. Building upon this foundation, Sec. 3.2 introduces our *contrasting and rethinking* procedure which embeds the dual-image comparison into a new reasoning path, guided by an LLM, to produce a more faithful rationale. The refined rationales are then used to construct a new reasoning dataset $\tilde{\mathcal{R}} = \{(v_i, q_i, a_i, \tilde{r}_i)\}_{i=1}^L$, which we name the **Vis**ual **Co**ntrastive **R**easoning dataset (VisCoR-55K). The VLM $\theta$ is updated to a new version $\tilde{\theta}$ with improved reasoning capability by finetuning on VisCoR-55K.

### 3.1 CONTRASTIVE VQA PAIR CURATION

To ensure the generalization of VC-STaR, the contrastive VQA pair curation pipeline should be flexible enough across a wide spectrum of VQA tasks. For better contrasting, each contrastive VQA pair $((v_i, q_i, a_i), (\hat{v}_i, \hat{q}_i, \hat{a}_i))$ should possess three key properties: (1) $q_i$ *and* $\hat{q}_i$ *are synonymous.* This shared question acts as a semantic anchor, grounding the two images $v_i$ and $\hat{v}_i$ at the same point in the semantic space. The images thus represent different manifestations of this anchor, providing a solid basis for contrasting; (2) $v_i$ *and* $\hat{v}_i$ *are visually similar.* $v_i$ and $\hat{v}_i$ should not be trivially distinct but exhibit visual similarity, creating a confusing contrasting. This visual proximity compels VLMs to engage in fine-grained contrasting to discriminate subtle differences; (3) $q_i$ *is reasoning dependent.* $q_i$ should be reasoning-provoking rather than one that can be solved by a straightforward answer. To achieve these requirements, as illustrated in Fig. 3, we propose a three-stage curation pipeline:

**Data Collection.** We collect 21 VQA datasets spanning five categories: reasoning (Zhang et al., 2019; Kiela et al., 2020; Lu et al., 2021b), graph/chart (Kembhavi et al., 2016; Mathew et al., 2022; Masry et al., 2022; Tang et al., 2023; Lu et al., 2023; Liu et al., 2024a), math (Lu et al., 2021a; Cao & Xiao, 2022; Gao et al., 2025), general (Zhu et al., 2016; Johnson et al., 2017; Acharya et al., 2019; Schwenk et al., 2022; Wang et al., 2023a; Chen et al., 2024b), and OCR (ICDAR, 2019; Yuan et al., 2022; Zhang et al., 2024b). This broad collection enriches the diversity of our curated pairs, which ensures the generalization ability of the finetuned model.

**Contrastive VQA Pair Hunting.** In order to compute the similarity of VQA pairs, we first represent the question $q_i$ and the image $v_i$ by high-dimensional embeddings, denoted as $e_i^q$ and $e_i^v$ respectively. We use GTE (Li et al., 2023c) text embeddings to represent the questions. In terms of image embedding, existing models fall into two types, *i.e.*, vision-language contrastive learning approaches (Radford et al., 2021; Tschannen et al., 2025) while vision-only self-supervised learning methods (Zhang et al., 2023; Oquab et al., 2024). The former ones mainly capture at global semantic information, while the later ones are good at instance discrimination. Neither of them are

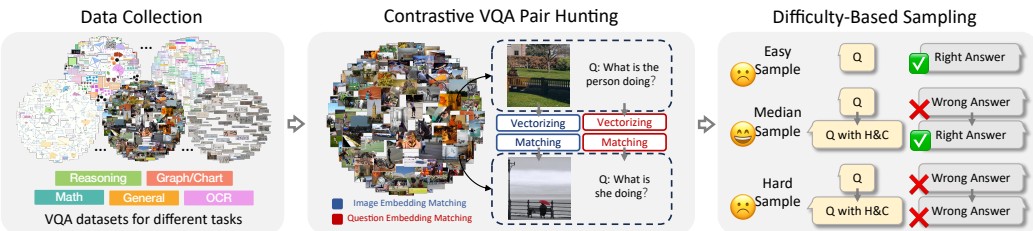

Figure 3: **Contrastive VQA pair curation pipeline.** To facilitate effective contrastive analysis, we curate corresponding challenging counterparts for VQA samples from a pool of diverse datasets. Each curated pair consists of two samples that share a synonymous question but feature distinct yet semantically similar images. Collected pairs are filtered by a difficulty-based sampling procedure.

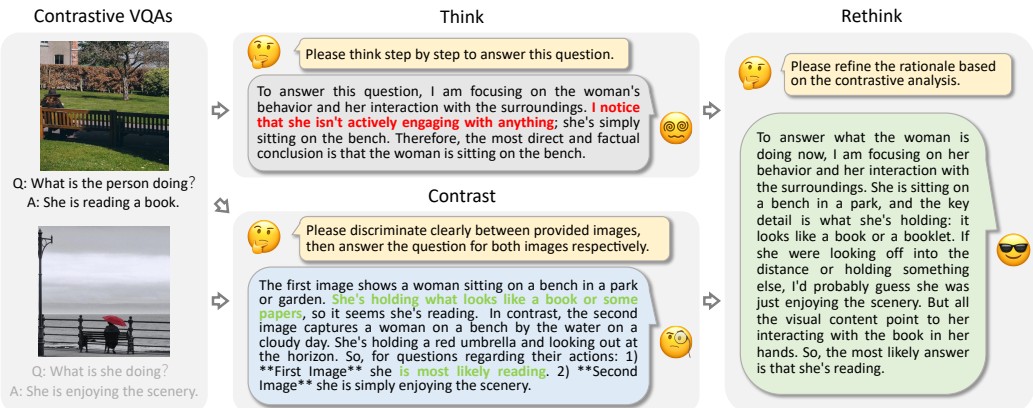

Figure 4: **Faithful rationale generation pipeline.** A contrastive analysis can be obtained based on the curated contrastive VQA pair. Leveraging the property of VLMs illustrated in Fig. 1, the contrastive analysis is then used to trigger a rethinking procedure, which refines the naive rationale into a more faithful one. This pipeline is designed to generate rationales for supervised finetuning.

generic enough to adapt to the diverse domains. To tackle the dilemma, we build a versatile visual embedding model based on ID-based visual metric learning (Ypsilantis et al., 2024; An et al., 2023). Hunting for a counterpart $(\hat{v}_i, \hat{q}_i, \hat{a}_i)$ is then performed dataset-by-dataset. A sample $(v_j, q_j, a_j)$ is recalled as a valid counterpart if it satisfies: $\gamma(e_i^v, e_j^v) < \phi_v$ and $\gamma(e_i^q, e_j^q) < \phi_q$, where $\gamma(\cdot, \cdot)$ is the cosine distance, and $\phi_v$ and $\phi_q$ are pre-defined thresholds for visual and question similarity, respectively. Any sample that fails to meet both conditions is dropped.

**Difficulty-Based Data Sampling.** For the goal of developing visual reasoning capability, $q_i$ should be a difficult question requires reasoning rather than a straightforward one. We define the levels of difficulty based on the performance of VLM $\theta$: (1) *easy samples* are with the simple $q_i$ which can be correctly answered by $\theta$ without any auxiliary help; (2) *median samples* are with $q_i$ which makes $\theta$ initially fails but succeeds when contrasting with $(\hat{v}_i, \hat{q}_i, \hat{a}_i)$ based on provided hint $a_i$ (the $C\&H$ setting introduced in Fig. 1); (3) *hard samples* are the ones with $q_i$ that cannot be correctly addressed by $\theta$ even with the help of contrasting. We only keep median-difficult contrastive VQA pairs for the rationale generating.

## 3.2 CONTRASTING AND RETHINKING

Rationales in the reasoning dataset $\mathcal{R} = \{(v_i, q_i, a_i, r_i)\}_{i=1}^M$ generated by the VLM $\theta$ itself include visual hallucinations. To achieve the goal:

$$\mathcal{R} = \{(v_i, q_i, a_i, r_i)\}_{i=1}^M \rightarrow \tilde{\mathcal{R}} = \{(v_i, q_i, a_i, \tilde{r}_i)\}_{i=1}^M, \tag{1}$$

where $\tilde{r}_i$ is the rectified rationale, we use the contrastive VQA counterpart $(\hat{v}_i, \hat{q}_i, \hat{a}_i)$ to provoke a rethinking action to refine $r_i$ into $\tilde{r}_i$. As illustrated in Fig. 4, this pipeline includes three steps:

**Thinking step.** Following the design of Zelikman et al. (2022) to provide the VLM $\theta$ with ground-truth answer $a_i$ as hints, we prompt the VLM $\theta$ to generate the coarse rationale $r_i$ for the target VQA

sample $(v_i, q_i, a_i)$ as follows:

$$r_i = f(v_i, q_i, a_i | \theta, \delta^t), \tag{2}$$

where $f$ is a inference process with a "thinking prompt" $\delta^t$. Details of $\delta^t$ are in the Sec. A.3.

**Contrasting step.** Asking the VLM $\theta$ to compare the target VQA sample $(v_i, q_i, a_i)$ with its contrastive counterpart $(\hat{v}_i, \hat{q}_i, \hat{a}_i)$ results in a contrastive analysis $c_i$ which may provide more faithful visual information:

$$c_i = f(((v_i, q_i, a_i), (\hat{v}_i, \hat{q}_i, \hat{a}_i)) | \theta, \delta^c), \tag{3}$$

where the $\delta^c$ is the "contrasting prompt". When $a_i$ has the same meaning as $\hat{a}_i$, $\delta^c$ requires summarizing the common patterns of $v_i$ and $\hat{v}_i$; When $a_i$ is different from $\hat{a}_i$, $\delta^c$ expects the analysis about the fine-grained differences between $v_i$ and $\hat{v}_i$. Details of $\delta^c$ are in the Sec. A.3.

**Rethinking step.** As demonstrated in Fig. 1, $c_i$ is more trustworthy than $r_i$. Hence, we adopt a LLM $\psi$ to transfer the information from $c_i$ to a new reasoning path according to $r_i$:

$$\tilde{r}_i = f(r_i, c_i | \psi, \delta^r), \tag{4}$$

where $\delta^r$ is the "rethinking prompt" which asks the LLM $\psi$ to rectify the visual hallucinations in $r_i$ according to the visual information from $c_i$. $\delta^r$ requires LLM $\psi$ to respond like directly answering the question $q_i$, details are in the Sec. A.3.

To ensure the quality of the $\tilde{\mathcal{R}}$, we finalize the visual reasoning dataset by employing a text-matching post-processing to filter out samples that contain incorrect reasoning patterns. The final visual reasoning dataset contains 55K VQA samples with corresponding rationales, *a.k.a.*, the VisCoR-55K.

## 4 EXPERIMENTS

Section 4.1 details our experimental setup, including the supervised finetuning process and the benchmarks used to evaluate the effectiveness of the VC-STaR. In Section 4.2, we present a comprehensive performance comparison. As a self-improving method for visual reasoning, we benchmark VC-STaR against two primary groups: (1) other self-improving baselines adaptable to visual reasoning, and (2) models trained on off-the-shelf visual reasoning datasets. Finally, Section 4.3 provides in-depth ablation studies on designs of our method, including the contrastive VQA pair construction, the generalization on other base models, the difficulty sampling strategy, and the effect of the types of contrastive VQA counterpart.

### 4.1 SETUP

**Implementation Details.** Using the LLaMA-factory framework (Zheng et al., 2024), we finetune the model for 3 epochs via full-parameter supervised finetuning (SFT), with the vision tower's parameters frozen. The SFT utilizes a learning rate of $1e$-5, a batch size of 256. The inference process of the finetuned model does not require such a contrastive pipeline illustrated in Fig. 4, and it follows the standard inference paradigm of VLMs. As for the curation of contrastive VQA pair, the question similarity threshold $\phi_q$ is set to 0.15 and the visual similarity threshold $\phi_v$ is set as 0.5 for datasets of general images. For the datasets including icon, geometry, chart or graph images, the visual similarity threshold $\phi_v$ is set as 0.3. The LLM $\psi$ used in the rethinking step of our rationale generation pipeline is the open-sourced Qwen2.5-72B.

**Evaluation Benchmarks.** We employ 6 benchmarks designed to assess its robustness against hallucination, mathematical reasoning, and general abilities. The MMVP (Tong et al., 2024) and Hallusion (Guan et al., 2024) benchmarks focus on visual hallucination, and the MathVista (Lu et al., 2024b) and MathVision (Wang et al., 2024) benchmarks are about the mathematical reasoning. The MMStar (Chen et al., 2024c) is a highly curated benchmark, composed of purified samples from multiple benchmarks, *e.g.*, MMMU (Yue et al., 2024) and MMBench Liu et al. (2024b). The MME-RealWorld benchmark Zhang et al. (2025b) is a large-scale, human-annotated benchmark for difficult, real-world tasks. Therefore, MMStar and MME-RealWorld are suitable to evaluate the general perceptual and cognitive abilities under varied scenarios.

Table 1: Performance comparison with self-improving baselines and the models trained on off-the-shelf visual reasoning datasets on hallucination, math, and general benchmarks. We adopt the Qwen2.5VL-7B as our base model, and report its reasoning performance as a baseline. MME-RW is short for MME-RealWorld Zhang et al. (2025b); R1-OV is short for R1-Onevision (Yang et al., 2025b). Blue (red) numbers in parentheses represent performance gains (drops) relative to the baseline. The best performance is in **boldface**, and the second best is underlined.

| Bench. Method | Hallucination | | Math | | General | | Avg. |
|---|---|---|---|---|---|---|---|
| | MMVP | Hallusion | MathVista | MathVision | MMStar | MME-RW | |
| Base Model | 70.0 | 53.1 | 68.4 | 24.0 | 61.8 | 55.9 | 55.5 |
| VQA SFT | 74.3(+4.3) | 54.2(+1.1) | 65.4(-3.0) | 19.4(-4.6) | 59.7(-2.1) | 56.8(+0.9) | 55.0(-0.5) |
| *Self-Improving Approaches* | | | | | | | |
| STaR(2022) | 73.0(+3.0) | 55.9(+2.8) | 66.9(-1.5) | 19.8(-4.2) | 58.9(-2.9) | 58.1(+2.2) | 55.4(-0.1) |
| Verifier(2024a) | 73.7(+3.7) | 53.2(+0.1) | 67.0(-1.4) | 20.3(-3.7) | 58.2(-3.6) | 56.7(+0.8) | 54.9(-0.6) |
| Feedback(2024) | 75.0(+5.0) | 53.4(+0.3) | 68.8(+0.4) | 22.1(-1.9) | 63.2(+1.4) | 56.0(+0.1) | 56.4(+0.9) |
| *Off-the-Shelf Visual Reasoning Datasets* | | | | | | | |
| Virgo(2025) | 68.0(-2.0) | 47.2(-5.9) | 63.5(-4.9) | 21.5(-2.5) | 59.7(-2.1) | 29.4(-26.5) | 48.2(-7.3) |
| LLaVA-CoT(2025) | 71.7(+1.7) | 50.3(-2.8) | 68.4(+0.0) | 24.4(+0.4) | 63.1(+1.3) | 59.3(+3.4) | 56.2(+0.7) |
| R1-OV(2025b) | 68.0(-2.0) | 55.8(+2.7) | 68.2(-0.2) | 25.4(+1.4) | 53.2(-8.6) | 46.3(-9.6) | 52.8(-2.7) |
| LPT(2025) | 74.0(+4.0) | 53.4(+0.3) | 69.2(+0.8) | 24.2(+0.2) | 64.3(+2.5) | 56.1(+0.2) | 56.9(+1.4) |
| VC-STaR(Ours) | 75.7(+5.7) | 56.3(+3.2) | 69.7(+1.3) | 25.3(+1.3) | 62.4(+0.6) | 59.3(+3.4) | 58.1(+2.6) |

## 4.2 MAIN RESULTS

**Comparison with the base model.** To evaluate the effectiveness of our approach, we employ Qwen2.5VL-7B as the base model and adopt the "think step by step" prompt to enable chain-of-thought reasoning. We compare our method against this baseline, with results summarized in Table 1. VC-STaR demonstrates consistent performance gains across diverse challenging benchmarks, achieving an average improvement of 2.4%. Notably, it yields substantial improvements of 5.7% and 3.2% on MMVP and the Hallusion Benchmark, respectively, validating its efficacy in mitigating hallucinations within the reasoning process. Our approach also shows its enhanced reasoning capabilities on mathematical benchmark, *i.e.*, MathVista and MathVision. Furthermore, the improvement on the MMStar and MME-RealWorld underscore the generalizability of the VC-STaR under varied challenging general-purpose scenarios.

For qualitative validation, Figure 5 provides visual comparisons that offer deeper insights. The visualizations reveal that our model excels at grounding its textual rationales in the corresponding visual evidence. This capability remains robust even when confronted with visually complex patterns, thereby effectively mitigating hallucinations.

**Comparison with self-improving baselines.** We reproduce three self-improving baselines and compare VC-STaR against them. Each baseline is applied to the Qwen2.5VL-7B base model and generates rationales on VisCoR-55K for finetuning, differing in their core improvement mechanism: (1) STaR (Zelikman et al., 2022): Leverages ground-truth answers to regenerate rationales for incorrect predictions. (2) Verifier (Lu et al., 2024a): Filters out visually hallucinated rationales via a self-verification step (Zhang et al., 2025a) to ensure visual grounding. (3) Feedback (Qu et al., 2024): Refines rationales based on self-generated feedback in a recursive manner. Table 1 reveals a critical trade-off: existing self-improving methods boost performance on hallucination benchmarks at the expense of math and general capabilities. Our approach mitigates this pattern and achieves robust, consistent performance gains.

**Comparison with off-the-shelf visual reasoning datasets.** We also evaluate VC-STaR against base model finetuned on four off-the-shelf visual reasoning datasets. These datasets represent diverse strategies for rationale generation. For instance, Virgo (Du et al., 2025) makes the VLM think slowly

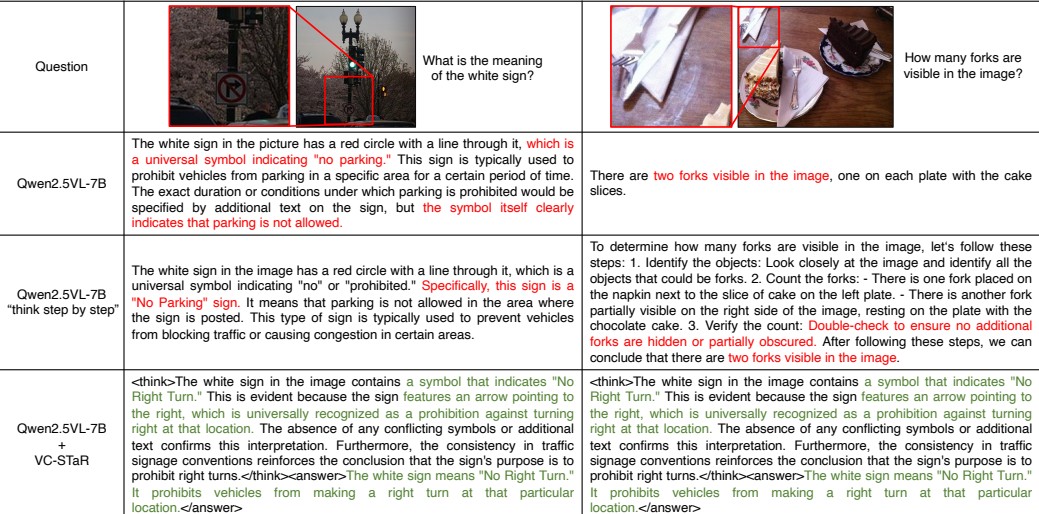

Figure 5: **Qualitative Comparison with base model.** The second row shows the directly response from the base model, the third row shows the response when the base model is prompted to "think stey by step", the last row shows the model improved with our VC-STaR. We highlight the key visual evidences with red boxes for clarity of visualization. More results are in Sec. A.4.

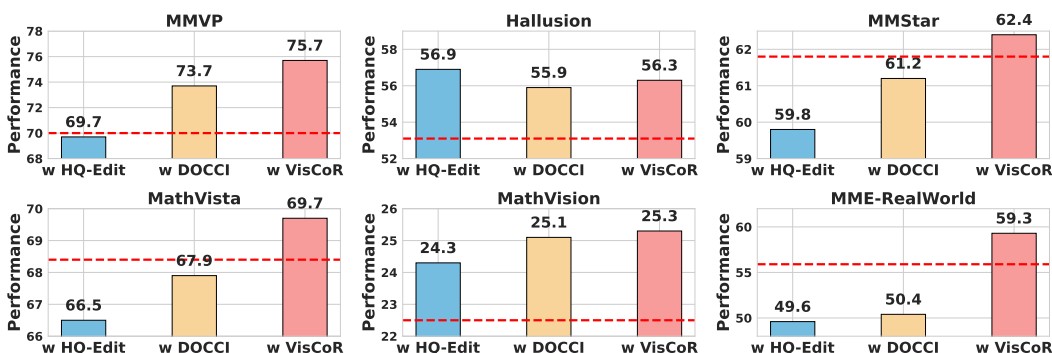

Figure 6: **Performance comparison with other contrastive VQA pair construction strategies.** Rationales in all settings are generated from the proposed VC-STaR. The red dashed line represents the base model (Qwen2.5VL-7B) performance.

with purely textual rationales. In contrast, LLaVA-CoT (Xu et al., 2025) leverages hand-crafted templates filled by the powerful GPT-4o (OpenAI, 2024a). Other approaches first convert visual information into text; R1-Onevision (Yang et al., 2025b) generates rationales from image captions using the DeepSeek-R1 model (Guo et al., 2025), while Long Perceptual Thought (LPT) (Liao et al., 2025) extends this by using dense captions (Onoe et al., 2024) and keywords like "wait" to elicit more detailed outputs from a similar LLM. In our experiments, we directly finetune the base model on each of these datasets. Based on the results shown in Table 1, we can draw the following conclusion: (a) Enhancing visual reasoning with purely textual rationales from Virgo is ineffective. This strongly indicates that visual modality matters. (b) The model trained on LLaVA-CoT suffers limited improvement, which demonstrates that the hand-crafted template struggle to generalize across diverse VQA tasks. (c) The models trained on datasets generated by DeepSeek-R1 based on captions achieves notable improvements. However, the performance gap between them and ours highlights the clear advantage of our visually-native approach over relying on textual captions.

### 4.3 ANALYSIS

**Can contrastive VQA pairs constructed in other ways?** To answer this, we explore alternative strategies for curating contrastive VQA pairs. The first strategy is editing-based, utilizing the HQ-

Table 2: Evaluation of the effect of VC-STaR on other base models. Blue numbers in parentheses represent performance gains.

| Model | VC-STaR | Hallusion | MathVision | MMStar |
|---|---|---|---|---|
| Qwen2.5VL | ✗ | 46.9 | 18.4 | 55.0 |
| 3B | ✔ | 53.2(+6.3) | 21.9(+3.5) | 55.7(+0.7) |
| InternVL2.5 | ✗ | 48.2 | 21.3 | 61.1 |
| 8B | ✔ | 55.4(+7.2) | 23.4(+2.1) | 62.5(+1.4) |

Table 3: Effect of the easy samples adding to VisCoR-55K. Red numbers in parentheses represent performance drops.

| $N_{easy}$ | Hallusion | MathVision | MMStar |
|---|---|---|---|
| 0k | 56.3 | 25.3 | 62.4 |
| +20k | 52.2(-4.1) | 23.3(-2.0) | 61.3(-1.1) |
| +40k | 55.7(-0.6) | 21.9(-3.4) | 59.5(-2.9) |

Table 4: Analysis about the effect of positive and negative contrastive VQA counterparts on GQA benchmark. We adopt the Qwen2.5VL-7B as our base model, and report its reasoning performance as a baseline. QR: query for relationships; QA: query for attributes; QG: query for global information; QC: query for category; CA: comparing of attribute; CC: choosing the object of one certain category; CAt: choosing the object of one certain attribute. Blue (red) numbers in parentheses represent performance gains (drops) relative to the baseline.

| Setting | Pos. | Neg. | QR | QA | QG | QC | CA | CC | CAt | Total |
|---|---|---|---|---|---|---|---|---|---|---|
| Base Model | - | - | 43.8 | 51.4 | 31.5 | 44.4 | 30.2 | 60.6 | 44.4 | 45.4 |
| VC-STaR | ✔ | ✗ | 48.3(+4.5) | 52.8(+1.4) | 46.8(+15.3) | 57.1(+12.7) | 42.9(+12.7) | 60.1(-0.5) | 44.4(+0.0) | 50.6(+5.2) |
| VC-STaR | ✗ | ✔ | 51.6(+7.8) | 56.8(+5.4) | 33.9(+2.4) | 59.2(+14.8) | 46.0(+15.8) | 70.8(+10.2) | 66.7(+22.3) | 53.7(+8.3) |
| VC-STaR | ✔ | ✔ | 53.5(+9.7) | 57.3(+5.9) | 46.3(+14.8) | 55.6(+11.2) | 36.5(+6.3) | 72.7(+12.1) | 55.6(+11.2) | 54.7(+9.3) |

Edit dataset (Hui et al., 2025). By prompting an LLM to create questions from editing instructions, we generate pairs where an original and an edited image yield different answers. The second strategy is caption-based, leveraging a dense caption dataset, *i.e.*DOCCI (Onoe et al., 2024). For this, we instruct an LLM to parse dense captions of visually similar images and generate a question that hinges on their subtle differences. For both strategies, we generate rationales for these newly created contrastive pairs using our proposed VC-STaR and finetune the Qwen2.5VL-7B. The results, presented in Fig. 6, lead to several observations: (a) VC-STaR is broadly effective, but performance is data-dependent. This is attributable to the biased data distribution of HQ-Edit and DOCCI, highlighting a key limitation of their curation scope. (b) VisCoR-55K includes contrastive pairs from a broader range of reasoning tasks, resulting in a more balanced performance.

**Does VC-STaR generalize to other base models?** We conduct experiments on Qwen2.5VL-3B and InternVL2.5-8B (Chen et al., 2025). Following the same self-improving procedure, we use VC-STaR to generate visual reasoning datasets from our VisCoR contrastive pairs, specifically for the two base models. We then finetune the Qwen2.5VL-3B and InternVL2.5-8B via the LLaMA-factory and SWIFT (Zhao et al., 2025), respectively. The results, presented in Table 2, demonstrate the model-agnostic effectiveness of our approach. These consistent and significant gains confirm that VC-STaR is a versatile and broadly applicable strategy for enhancing the visual reasoning ability.

**What is the effect of easy samples on visual reasoning?** Starting with our VisCoR-55K datasets, we incrementally add easy samples of two batches with 20K each. As illustrated in Table 3, we observe that the inclusion of easy samples is harmful. Specifically, when the number of easy samples increases, performance decreases. Therefore, we do not use the easy samples to avoid the potential "overthink" for straightforward problems.

**How the contrastive VQA pairs of different types contribute?** A contrastive VQA pair can be categorized as "positive" if both samples yield the same answer, and "negative" if their answers differ. To investigate the respective contributions of these two types of counterparts to our method's performance, we conducted a controlled experiment on the GQA dataset (Hudson & Manning, 2019). The structured nature of GQA allows for the reliable curation of both positive and negative pairs via simple text matching. We applied VC-STaR to three distinct training sets: one generated from only positive contrastive pairs, one from only negative pairs, and a combined set including both. The results, detailed in Table 4, reveal a clear and significant trend. While both types of pairs are beneficial, negative counterparts are substantially more effective than positive ones, and their combination

yields the optimal total gain, highlighting their complementary roles. We attribute the superior efficacy of negative counterparts to their ability to induce stronger semantic contrast. Accordingly, our approach incorporates both positive and negative pairs without restriction to achieve optimal gain.

## 5 CONCLUSION

We demonstrate that visual hallucination can be effectively mitigated through the lens of contrast, thereby enhancing visual reasoning. Based on the insight that VLMs can see better by contrast, we propose the VC-STaR. The VC-STaR refines hallucinatory reasoning paths through analysis over curated contrastive VQA pairs, which yields our high-quality VisCoR-55K. Finetuning on VisCoR-55K delivers a consistent performance gain across six benchmarks, significantly surpassing other self-improving baselines and models trained on state-of-the-art visual reasoning datasets. Looking forward, we hope our work will offer a new perspective on visual reasoning and inspire the exploration of novel contrast-driven training and inference paradigms.

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

## A APPENDIX

### A.1 RETHINKING VC-STAR FROM A COGNITIVE PERSPECTIVE

Learning and reasoning are inherently comparative and contrastive processes. Humans rarely learn concepts in isolation. Instead, humans refine our understanding by comparing examples, identifying distinguishing features, and reasoning through analogies and differences. The prototype theory concludes this cognitive behavior as that our human beings identify new identities by comparing them with the prototype concept (Rosch, 1975). Besides, the structure-mapping theory says that analogical reasoning can recognize the relationships shared by two domains (Gentner, 1983). This mapping can be treated as a fine-grained contrasting process. In our work, the contrasting process provides an opportunity to learn visual concept from the prototype, and our rethinking strategy reinforces the structure-mapping by generating new reasoning paths via contrasting. We hope to highlight the potential of porting such human-like cognitive behaviors to the domain of reasoning.

### A.2 DETAILS ABOUT VISCOR-55K

The construction of our VisCoR-55K dataset is a multi-stage process involving efficient pair curation, difficulty-based filtering, and quality-controlled rationale generation. The entire pipeline is designed to produce a high-quality, challenging visual reasoning dataset. Our curation process for contrastive VQA pairs begins with a dataset-by-dataset, divide-and-conquer strategy. To maintain computational tractability and avoid a costly $O(n^2)$ search complexity across the entire data pool, we implement a greedy, first-match-exit search algorithm: for each sample within a given source dataset, the search for a contrastive VQA counterpart terminates as soon as the first valid match is identified. This efficient approach allows us to scale the curation process effectively. Following this

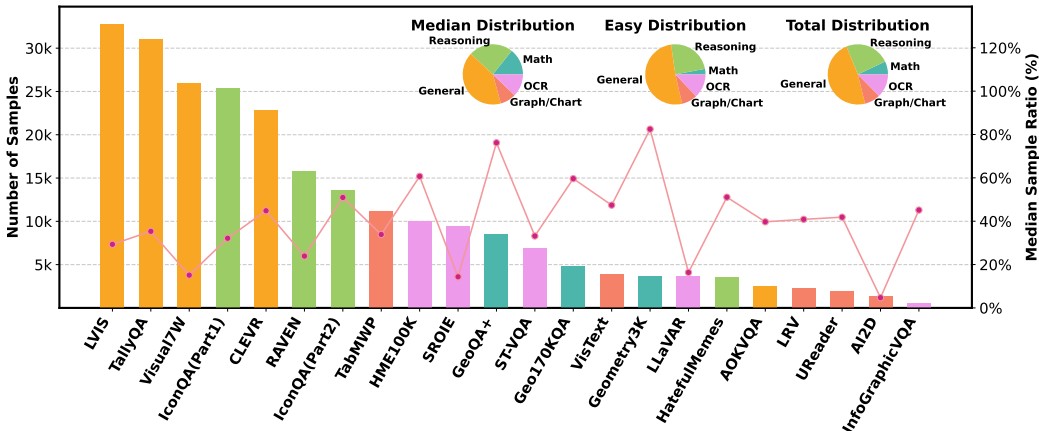

Figure 7: **Statistics of the contrastive VQA pair curation.** The bar chart (left y-axis) displays the total number of contrastive VQA pairs in each dataset, with colors indicating the data category. The line graph (right y-axis) plots the ratio of median samples identified within those pairs for each dataset. In the upper right, the pie charts provide a categorical breakdown of the sample.

procedure, we initially curated a large pool of 240k raw contrastive VQA pairs. The distribution is visualized by the bar chart in Fig. 7, with the left y-axis indicating the number of samples.

This initial pool of 240k pairs then undergoes a rigorous filtering and refinement pipeline. First, we apply the difficulty-based sampling strategy (as detailed in Sec. 3.1) to select only the median samples, which are most effective for enhancing the model's reasoning capabilities. The proportion of median samples varies significantly across datasets, and is illustrated by the line graph in Fig. 7 (plotted against the right y-axis). This critical filtering step narrows our collection down to 86k challenging contrastive pairs. Subsequently, we leverage the contrasting and rethinking pipeline to generate a high-quality rationale for each of these 86k samples. As a final quality control measure, we employ a text-matching-based post-processing step to automatically filter out any rationales containing unexpected or erroneous reasoning patterns. This process culminates in our final VisCoR-55K dataset, a collection of high-fidelity visual reasoning samples ready for finetuning. The pie charts in Fig. 7 provide a categorical overview of the data composition throughout this pipeline.

### A.3    PROMPTS FOR THINKING, CONTRASTING, AND RETHINKING

As introduced in Sec. 3.2, 3 steps lead to the final rationales. We design 3 prompts for the thinking, contrasting, and rethinking steps. The thinking prompt is:

```
You are a helpful assistant to answer the question by thinking step by step.
### INPUT ###
- Image: The image that serves as the basis for answering the question.
- Question: The question pertains to the content of the image.
- Answer: The correct answer for the question about the image.
### INSTRUCTION ###
- You should analyze the question and decide to focus on which visual content.
- You should parse the details of visual content based on the question.
- You should conclude the visual evidence to answer the question.
### OUTPUT ###
- The returned content MUST be in the natural flow.
<Image><Question><Answer>
```

The contrasting prompt is:

> You are a helpful assistant to think step by step for discriminating between two images to answer two synonymous questions.
> ### INPUT ###
> - First Image: One image that serves as the basis for answering the question.
> - Second Image: The other image that serves as the basis for answering the question.
> - First Question: The question pertains to the content of the First Image.
> - Second Question: The question pertains to the content of the Second Image.
> - Answer: The correct answer for the question about the images.
> ### INSTRUCTION ###
> - When the correct answers for the two images are the same, you should summarize the common patterns in the visual content of the two images.
> - When the correct answers for the two images are different, you should identify the differences in visual content between two images.
> - Conclude the visual evidence to answer the questions respectively.
> ### OUTPUT ###
> - Return in the natural flow.
> `<FirstImage><SecondImage>`
> `<FirstQuestion><SecondQuestion><Answer>`

The "Answer" here is the concatenation of both samples. The rethinking prompt is:

> You are a helpful assistant to rewrite the coarse rationale into a more correct and more logical one based on a contrastive analysis.
> ### INPUT ###
> - Question: The question to be answered based one given target image.
> - Answer: The correct answer to answer the question.
> - Coarse Rationale: The naive reasoning process answering the question.
> - Contrastive Analysis: The reasoning process when comparing the first image with the second image for synonymous questions.
> ### INSTRUCTION ###
> - The contrastive analysis is more reliable than the coarse rationale.
> - If the answers in the contrastive analysis are the same for the two images, the model should formulate a summary reasoning schema. This schema must summarize the key visual features and confirm that the provided visual evidence aligns with this schema to derive the conclusion.
> - If the answers in the contrastive analysis are different for the two images, you can employ backward chaining hypothesizing the visual cues that would be present if the alternative answer were correct, and then highlighting the critical distinctions between this hypothetical scenario and the actual visual evidence.
> ### OUTPUT ###
> - The output MUST be in the format of '`<think>`the thinking content`</think><answer>`the answering content`</answer>`'.
> - The content of thinking content MUST be between the special token of '`<think>`' and '`</think>`'
> - The content of answering content MUST be between the special token of '`<answer>`' and '`</answer>`'.
> `<Question><Answer><CoarseRationale><ContrastiveAnalysis>`

### A.4 ADDITIONAL QUALITATIVE RESULTS

Examples of rationales generated by VC-STaR in VisCoR-55K are illustrated in Fig. 8. After obtaining the model finetuned with VisCoR-55K on Qwen2.5VL-7B, we test it on some customized visual question answering cases and observe the interesting results shown in Fig. 9.

| Question | Rationales |
|---|---|
|  Choose the figure that would logically complete the sequence. | \<think>To determine the figure that logically completes the sequence, let's carefully analyze the target image: 1. **Shape Analysis**: - The shapes in the sequence are pentagons, and they remain consistent in form across all positions. 2. **Size Progression**: - Observing the figures, there is a clear trend of increasing size as the sequence progresses from left to right and top to bottom. Each subsequent shape grows larger than the previous one. 3. **Color/Pattern Observation**: - There is no apparent change in color or shading patterns influencing the placement of the figures. The consistency in color suggests that size is the primary factor driving the sequence. 4. **Conclusion Based on Size Increase**: - Given the established pattern of gradual size expansion, the next figure in the sequence must continue this trend by being larger than the preceding shapes while maintaining the same form. The details in the image confirm that the progression relies solely on incremental growth in size without introducing additional complexities such as color changes or new shapes. Therefore, the figure that aligns perfectly with this principle is option D.\</think>\<answer>D\</answer> |
|  How many buses are there? | \<think>The image shows two red London United buses parked side by side, with their front ends visible. Therefore, based on this observation: There are **two** buses shown in the picture.\</think>\<answer>2.\</answer> |
|  How many horns are visible? | \<think>The image shows an animal positioned in such a way that only one horn is clearly visible, as the rest of its head is partially obscured by an archway. This indicates that while there may be another horn present, it is not visible from this angle or due to the obstruction. The visual evidence supports the conclusion that only one horn is identifiable in the given view.\</think>\<answer>1\</answer> |
|  In the given diagram, triangle ABC is inscribed within circle O, and D is the midpoint of side BC. If OD is extended to intersect circle O at point E, and EC is then connected, what is the measure of angle A if angle OEC is 65°? Choices: A: 50° B: 55° C: 60° D: 65° | \<think>To determine the measure of angle A, we start by analyzing the given information and leveraging geometric properties. Triangle ABC is inscribed in circle O, with D as the midpoint of side BC. Since D is the midpoint, OD is perpendicular to BC, and OD extended intersects the circle at point E. Connecting EC, we are given that angle OEC is 65°. From the property of isosceles triangles, since OE and OC are radii of the circle, triangle OEC is isosceles, and thus angles OEC and OCE are equal. Therefore, angle EOC can be calculated as: \\[ \text{Angle EOC} = 180^\circ - 2 \times 65^\circ = 50^\circ. \\] Next, because D is the midpoint of BC and OD is perpendicular to BC, it follows that CE equals BE. This symmetry implies that angle EOB is also 50°. Consequently, the central angle BOC, which subtends arc BC, is twice angle EOB: \\[ \text{Angle BOC} = 2 \times 50^\circ = 100^\circ. \\] Finally, angle A, being an inscribed angle subtending the same arc BC, is half of the central angle BOC: \\[ \text{Angle A} = \frac{1}{2} \times 100^\circ = 50^\circ. \\] This reasoning aligns with the specific details provided in the problem and confirms the measure of angle A.\</think> \<answer>A: 50°\</answer> |

Figure 8: **Examples of rationales in VisCoR-55K.**

| Question | Qwen2.5VL-7B | Qwen2.5VL-7B "think step by step" | Qwen2.5VL-7B+VC-STaR |
|---|---|---|---|
|  What makes the people so shocked? | The image shows a group of people in what appears to be a sports stadium, reacting with shock and surprise. The person in the center is holding a baseball bat, which could suggest that something unexpected or dramatic has occurred on the field, such as a foul ball hitting someone or a player making an extraordinary play. The expressions and body language of the spectators indicate a sudden and intense reaction to an event that has just taken place. Without additional context, it's not possible to determine the exact cause of their shock, but it is clear that whatever happened was surprising and possibly dangerous. | To determine what makes the people in the image so shocked, let's analyze the context and details: 1. Setting: The image appears to be taken at a baseball game, as indicated by the presence of a baseball bat and the crowd's attire, which includes team colors and logos. 2. Action: One person is holding a baseball bat, which suggests that something unexpected or dramatic has occurred on the field. This could be a play that resulted in a controversial or surprising outcome, such as a home run, a foul ball, or an unusual play. 3. Reactions: The people in the image are displaying strong reactions—some are covering their faces with their hands, while others are looking towards the field with expressions of shock or surprise. This indicates that whatever happened was unexpected and likely had a significant impact on the game. 4. Contextual Clues: The fact that the bat is being held up suggests that it might have been used in a way that surprised the spectators, possibly due to a foul ball or a hit that went into the stands, leading to a potential injury or disruption. Given these observations, the people are likely shocked because of a surprising event on the field, such as a foul ball hitting someone in the stands, a controversial play, or an unexpected turn of events during the game. The exact cause would depend on the specific circumstances of the game, but the general reaction points to a moment of high drama or surprise. | \<think>The image depicts a scene at what appears to be a baseball game, with spectators showing expressions of surprise or shock. The key detail is the presence of a bat in mid-air, which suggests an unexpected event such as a foul ball or a player's swing. This plays with the idea that something sudden and surprising has occurred during the game, causing the audience's reaction. While other scenarios might involve different causes for shock—such as accidents or unrelated incidents—the visual evidence here points directly to the bat as the source of their astonishment.\</think>\<answer>The people are shocked because a bat has been thrown into the crowd, likely during a moment of excitement or surprise at the baseball game.\</answer> |
|  What color is the car on the parking spot 33? | The car parked in the parking spot labeled "33" is blue. | The car on the parking spot labeled "33" is blue. | \<think>The question asks about the color of the car in parking spot 33. The analysis shows that there is no car occupying parking spot 33 in the image, as it appears empty. Since there is no vehicle present in this specific spot, it is impossible to determine any color associated with it. This conclusion aligns with the observation that the absence of a car directly answers the question.\</think> \<answer> There is no car parked in parking spot 33, so I cannot determine its color.\</answer> |
|  What answer will the girl with a single braid write on the board? | The girl with a single braid is working on the problem "11 − 2". The answer to this problem is 9. | To determine what answer the girl with a single braid will written on the board, we need to solve the mathematical expression she is working on: The expression on the board for the girl with a single braid is : 7×2. Let's break it down step by step: 1. Identify the operation: the operation here is multiplication. 2. Multiply the numbers: 7 multiplied by 2 is 14. So, the girl with a single braid will write the number 14 on the board. | \<think>To answer the question "What answer will the girl with a single braid write on the board?", let's break down the analysis step by step: 1. Identify the Girl with Single Braid: - The girl with a single braid is located on the far left side of the image. 2. Position on the Board: - She is positioned directly below the equation (3 \times 3 =). 3. Action with Chalk: - The girl is holding a piece of chalk, indicating that she is ready to write. 4. Conclusion from Visual Evidence: The equation (3 (times 3) equals 9. The girl is positioned directly below this equation, suggesting she will write the result. Based on these observations, it is clear that the girl with a single braid will write the number 9 on the board as the solution to the equation (3 \times 3 =). Therefore, the answer is: The girl with a single braid will write the number 9 on the board.\</think>\<answer> The girl with a single braid will write the number 9 on the board.\</answer> |

Figure 9: **Additional qualitative comparison.**

