# OpenReview forum: "Through the Lens of Contrast: Self-Improving Visual Reasoning in VLMs"
_ICLR.cc/2026/Conference — ICLR 2026 Oral_

### Official Review · Reviewer_4FzS · 2025-10-26

**Soundness:** 3
**Presentation:** 3
**Contribution:** 3
**Rating:** 6
**Confidence:** 4

**Summary:**

This paper introduces Visual Contrastive Self-Taught Reasoner (VC-STaR), a self-improving visual reasoning framework. The key insight is that VLMs can identify more relevant visual cues when comparing two visually similar images paired with synonymous questions. Building on this observation, the authors collect visually similar image–question pairs and generate rationales for self-improvement training. Experiments across multiple benchmarks demonstrate that the contrastive capability of VLMs can effectively bootstrap their own visual reasoning ability.

**Strengths:**

1. The method is straightforward. By leveraging the intrinsic contrastive ability of VLMs, VC-STaR enables the model to generate improved rationales through self-prompting, making it readily adaptable to new architectures and settings.

2. VC-STaR achieves substantial performance gains, surpassing prior self-improvement methods and even models fine-tuned on carefully curated datasets. Moreover, the method generalizes well across different backbone models, further validating its effectiveness and robustness.

**Weaknesses:**

1. Construct data in a contrastive manner is not a new idea, there are several previous works already explored training VLMs with collected contrastive samples, such as [1] and [2]. I suggest authors to include a discussion with these works for a better understanding with training VLMs using contrastive data samples.

    [1] Jiao et al. Img-Diff: Contrastive Data Synthesis for Multimodal Large Language Models

    [2] Ma et al. C3L: Content Correlated Vision-Language Instruction Tuning Data Generation
    via Contrastive Learning

2. Despite the effectiveness, the proposed VC-STaR framework appears to be significantly more computationally expensive than existing methods such as STaR. Pairing visually similar samples from a large dataset may hinder the method’s scalability and efficiency when adapting to new domains. A discussion or analysis of VC-STaR’s efficiency relative to previous methods would strengthen the paper.

3. Table 1 does not include a baseline that trains models solely with instruction–response pairs. Including such a baseline would help readers assess the specific contribution of rationales and better understand the overall performance gain.

4. The paper does not analyze how the quality of the contrastively selected samples affects performance. Since these samples are central to the proposed pipeline, a discussion or empirical analysis of their quality’s influence on model outcomes would be valuable.

**Questions:**

1. Can the authors provide a quantitative or qualitative comparison of VC-STaR’s computational cost relative to STaR?
2. How sensitive is VC-STaR to noisy or low-quality contrastive samples?

---

> ### Author Response · Authors · 2025-11-21
> **Response for Reviewer 4FzS Part 1**
>
> We sincerely thank the reviewer for their valuable and detailed feedback. We are encouraged that the reviewer recognized our method as "straightforward" and effective, with "substantial performance gains."
>
> > Construct data in a contrastive manner is not a new idea, there are several previous works already explored training VLMs with collected contrastive samples, such as [1] and [2]. I suggest authors to include a discussion with these works for a better understanding with training VLMs using contrastive data samples.
>
> We sincerely thank the reviewer for this critical question and for pointing out these highly relevant works (Img-Diff, C3L). We agree that a discussion is essential to properly position our contribution. While these papers and ours all leverage "contrastive" principles, we argue that they do so for fundamentally different purposes and through different mechanisms. In short, prior works use contrast to create new instruction-tuning data for a comparative task, whereas we use contrast as a scaffolding mechanism to enable a reasoning self-correction loop for a single-image task. To the best of our knowledge, we are the first to study the benefit of contrasting for visual reasoning.
>
> To elaborate, the works cited by the reviewer focus on generating or refining instruction-following datasets: Img-Diff generates pairs of similar images with object variations and then creates question-answer pairs that explicitly ask the model to describe the difference. The final training data teaches the model to perform the task of comparing two images; C3L uses a contrastive learning module to improve a data generator. It distinguishes between image-relevant ("positive") and image-irrelevant ("negative") generated instructions, training the model to produce better-grounded VQA data. The contrast here serves to filter and refine the quality of the instruction data itself. In stark contrast, our VC-STaR framework uses the contrastive pair for a different purpose: as the basis of self-correction within a reasoning process. We will add the discussion about Img-Diff and C3L in the Power of Contrasting paragraph (line 155).
>
> >Despite the effectiveness, the proposed VC-STaR framework appears to be significantly more computationally expensive than existing methods such as STaR. Pairing visually similar samples from a large dataset may hinder the method’s scalability and efficiency when adapting to new domains. A discussion or analysis of VC-STaR’s efficiency relative to previous methods would strengthen the paper.
>
> We thank the reviewer for this insightful comment and agree that reporting resource usage is crucial. Our approach introduces a modest cost in data generation, while maintaining a fine-tuning cost comparable to existing self-improving frameworks like STaR.
>
> - Fine-tuning Cost: We conduct standard Supervised Fine-Tuning (SFT) on the 7B model using the LLaMAFactory framework once the VisCoR-55K dataset is generated. This process takes approximately four hours on a single node with 8x A800 GPUs. This cost is nearly identical to the standard fine-tuning procedure in approach like STaR.
> - Data Generation Cost: The additional cost stems from our data generation pipeline. The one-time contrastive VQA pair mining step (L238-L251) processes approximately 7 samples per second；The 'rethink' step (L281-L290) generates reasoning paths at about 1 sample per second. We argue this is a reasonable and acceptable investment to create a large-scale, high-quality reasoning dataset. To relieve this generation cost for the community and foster future research, we have committed to releasing the full VisCoR-55K dataset, including the intermediate contrastive VQA pairs.
>
> Furthermore, we have already implemented optimizations to accelerate the generation, including multi-processing concurrency, batch-wise computation for pair mining in a large-scale raw dataset, and leveraging the vLLM framework for fast inference. While our current implementation is effective, we believe there is still room for further engineering optimizations to improve efficiency for further scaling up. We will add the discussion about the resource usage into our paper.

---

> > ### Author Response · Authors · 2025-11-21
> > **Response for Reviewer 4FzS Part 2**
> >
> > >Table 1 does not include a baseline that trains models solely with instruction–response pairs. Including such a baseline would help readers assess the specific contribution of rationales and better understand the overall performance gain.
> >
> > Thanks for your advise, we conduct this experiment and get the following results:
> > | | MMVP | Hallusioin | MathVista | MathVision | MMStar | Avg. |
> > |:---|:---:|:---:|:---:|:---:|:---:|:---:|
> > | Base Model | 70.0 | 53.1 | 68.4 | 24.0 | 61.8 | 55.5 |
> > | Instruction SFT | 74.3 | 54.2 | 65.4 | 19.4 | 59.7 | 54.6 |
> > | Reasoning | 75.7 | 56.3 | 69.7 | 25.3 | 62.4 | 57.9 |
> >
> > We will add the results into the Table 1 of our paper.
> >
> > >The paper does not analyze how the quality of the contrastively selected samples affects performance. Since these samples are central to the proposed pipeline, a discussion or empirical analysis of their quality’s influence on model outcomes would be valuable.
> >
> > The reviewer asks about the correctness of the answer for the second, contrasted image. This is a very important point. At first, to avoid misunderstanding, we need to clarify: In our framework, the contrastive counterparts are sampled directly from widely-used public VQA datasets. We operate under the assumption that the annotations in these established datasets are, for the most part, reliable and correct.
> >
> > However, we have considered the quality of the contrastive counterparts by analyzing "how contrastive VQA pairs constructed in other ways?". Results of which are shown in Fig.6. We test this from two perspectives: QA quality and data diversity.
> > - QA Quality: When we conducted VC-STaR on the DOCCI dataset, the Question-Answer pairs for the contrastive samples were automatically generated from dense captions. In this setting, the quality and correctness of the QA pairs cannot be guaranteed to the same extent as in human-annotated datasets. As discussed in our original analysis of Figure 6, the results on DOCCI showed more limited improvement precisely because the lower quality of these machine-generated QA pairs introduced noise into the contrastive learning process.
> > - Diversity: When we conducted VC-STaR on the HQ-Editing dataset, our experiments with the HQ-Editing dataset revealed its inherent limitations. As the dataset is restricted to perception-level contrasts derived from editing prompts, its diversity is narrow. This resulted in performance gains exclusively on the Hallusion Bench, while causing performance degradation across all other benchmarks.
> >
> > In summary our results demonstrate that the reliability of QA and the diversity of data is important. We hope these results can address the concerns of the reviewer.
> >
> > ---
> > We sincerely apologize, but due to time constraints, we have not yet revised the PDF document. We will update the manuscript as soon as possible and will leave a comment to guide the reviewer to the changes once it is uploaded.

---

> > > ### Comment · Reviewer_4FzS · 2025-11-24
> > >
> > > Thank you for your detailed rebuttal. The responses address my previous concerns, and I am now confident in the soundness and contribution of the work. I am therefore raising my overall score to 8.

---

> > > > ### Author Response · Authors · 2025-11-24
> > > >
> > > > We are particularly encouraged that our rebuttal successfully addressed your concerns. Your insightful comments have been invaluable in improving our paper, and we truly appreciate your support.

---

### Official Review · Reviewer_mDXQ · 2025-10-30

**Soundness:** 3
**Presentation:** 3
**Contribution:** 3
**Rating:** 8
**Confidence:** 3

**Summary:**

This paper introduces VC-STaR, a contrastive self-improvement framework that reduces visual hallucinations in vision–language models by pairing similar images with synonymous questions and applying a three-stage process (coarse reasoning, contrastive analysis, rationale rewriting). Using this pipeline, the authors build VisCoR-55K from 21 VQA datasets and fine-tune models, yielding consistent gains on hallucination, math, and general benchmarks over STaR-like baselines and text-only rationale training.

**Strengths:**

1. Novel and Elegant Methodology: The core insight—that visual contrast mitigates hallucinations—is intuitive yet effective. The proposed three-step VC-STaR framework is a simple, "visually-native" self-improvement pipeline that avoids complex external components like reward models.

2. Systematic Data Contribution: The paper introduces VisCoR-55K, a high-quality visual reasoning dataset created through a scalable and principled curation pipeline. It also provides valuable design insights, such as the superior efficacy of "negative" (different-answer) contrastive pairs.

3. Broad and Robust Empirical Validation: The approach achieves consistent and significant performance gains across a diverse set of challenging benchmarks (hallucination, math, general reasoning). Its effectiveness is demonstrated across multiple base models, showing strong generalization and outperforming key self-improvement baselines.

**Weaknesses:**

1. Some key details could be clarified. For example, how the similarity thresholds (φv/φq) were chosen and their sensitivity, as well as the configuration/source of the LLM (ψ) used in the rethinking stage. Adding these would help with reproducibility and understanding the method’s scope.

2. The evaluation coverage feels a bit limited. Including results on benchmarks such as POPE, MMBench, MME-RealWorld, TallyQA, and LLaVA-Eval would provide a more complete picture of performance across settings.

**Questions:**

1. Visual Similarity Computation
Please provide details on how visual similarity is computed, including the architecture, training data, and training objectives of the “general visual embedding model.” Indicate whether the model reuses or fine-tunes existing models (e.g., CLIP, DINOv2, UNICOM, UDON), and include specific implementation details.

2. Similarity Threshold Selection
Please explain how the text and visual similarity thresholds (φq and φv) are determined and provide corresponding sensitivity analyses.

3. LLM (ψ) in the “Rethinking” Stage
Please clarify the relationship between ψ and the vision-language model θ: is ψ identical to θ, or is it a stronger external LLM? If the latter, please describe its role and contribution to performance improvements.

4. Additional Benchmark Evaluations
Please include comprehensive evaluation results on more benchmarks (e.g., POPE (hallucination), MMBench, MME-RealWorld, TallyQA, LLaVA-Eval) to provide a more complete view of the model’s performance.

---

> ### Author Response · Authors · 2025-11-21
> **Response for Reviewer mDXQ Part 1**
>
> We are deeply grateful to the reviewer for their positive and insightful feedback, and for the high rating of our work. We try to address your concerns one-by-one:
>
> >Some key details could be clarified. For example, how the similarity thresholds (φv/φq) were chosen and their sensitivity, as well as the configuration/source of the LLM (ψ) used in the rethinking stage. Adding these would help with reproducibility and understanding the method’s scope.
>
> Regarding the similarity thresholds, these were determined empirically during the initial phase of our work. We started with a small, diverse batch of data and adjusted the thresholds while visually inspecting the resulting contrastive pairs. During this adjusting, we observed a threshold of 0.5 was effective for obtaining stable contrastive pairs from natural images. In contrast, for structured images like charts and diagrams, which possess a higher degree of inherent similarity, our inspection revealed that a stricter threshold of 0.3 was necessary. We acknowledge the reviewer's point about sensitivity analysis; we opted for this small-scale, visual confirmation because a full, closed-loop analysis, which would require re-running the entire data generation and model fine-tuning pipeline for each threshold adjustment, is prohibitively expensive. Encouragingly, the thresholds determined from this small-scale validation proved highly effective during large-scale data construction, consistently producing meaningful and interesting contrastive pairs (as illustrated in Figure 2 of our paper). This suggests that our framework is robust to the precise choice of these thresholds and that they do not require delicate hyperparameter tuning. The process for the text similarity threshold was more straightforward; following a similar validation process, we were able to establish a single, unified threshold that worked effectively across all datasets.
>
> Regarding the LLM used in the "rethink" stage, we utilized the open-sourced Qwen2.5-72B model. It is crucial to clarify its role: as shown in the prompt, we use this powerful LLM strictly as a rewriter. Its task is solely to fuse the insights from our contrastive analysis into a coherent  reasoning path for the anchor image. We donot inject its own world knowledge into reasoning paths, but to require it act as a high-fidelity text-editing tool. This practice of using a capable LLM as a constrained "rewriter" or "synthesizer" to generate training data is a common and effective strategy, also seen in other contemporary works such as R1-Onevision. This ensures the reasoning is grounded in our model's analysis, not the rewriter's knowledge.
>
> >The evaluation coverage feels a bit limited. Including results on benchmarks such as POPE, MMBench, MME-RealWorld, TallyQA, and LLaVA-Eval would provide a more complete picture of performance across settings.
>
> We completely agree that a more comprehensive assessment provides a more complete picture of our model's performance. This feedback also aligns with comments from other reviewers (EV9r and RMKH) who expressed a particular interest in expanding the evaluation on general-purpose benchmarks. Given the limited time and resources during the rebuttal period, we had to strategically prioritize which of the valuable suggested benchmarks to address first.
>
> We chose to focus our efforts on MME-RealWorld (MME-RW) for the reason: MMStar, which is already a core part of our evaluation, is a highly curated benchmark that incorporates the gold samples from several other benchmarks, including MMBench. To avoid redundancy and to test our model against a different set of challenges, we opted for MME-Realworld. MME-Realworld is a newer, large-scale, and notably difficult benchmark, and we believe demonstrating strong performance here is of significant importance. Due to the limited rebuttal period, we have initially tested the performance of VC-STaR (59.3), where it achieves a +3.4 point improvement over the base model (55.9). We are committed to testing the other baselines on MME-Realworld and will update Table 1 accordingly for the final version of the paper. This will ensure that all three categories in our main results table are robustly supported by two distinct benchmarks.

---

> > ### Author Response · Authors · 2025-11-21
> > **Response for Reviewer mDXQ Part 2**
> >
> > >Visual Similarity Computation Please provide details on how visual similarity is computed, including the architecture, training data, and training objectives of the “general visual embedding model.” Indicate whether the model reuses or fine-tunes existing models (e.g., CLIP, DINOv2, UNICOM, UDON), and include specific implementation details.
> >
> > Our model is trained from UNICOM, following similar process illustrated in the "Fine-tune for Supervised Image Retrieval." section in the UNICOM paper. A key feature of our approach is the integration of instance-level data, which equips our model to discern both global-level and instance-level similarities. We are fully aware that this custom component may raise valid concerns about reproducibility. To address this, we make a commitment to ensure the embedding model is publicly accessible immediately following the paper's acceptance.
> >
> > ---
> >
> > We sincerely apologize, but due to time constraints, we have not yet revised the PDF document. We will update the manuscript as soon as possible and will leave a comment to guide the reviewer to the changes once it is uploaded.

---

### Official Review · Reviewer_EeLq · 2025-10-31

**Soundness:** 3
**Presentation:** 3
**Contribution:** 2
**Rating:** 6
**Confidence:** 3

**Summary:**

Motivated by an observation that VLM performs better when presented with a contrastive pair of VQA, the authors propose a “Visual Contrastive Self-Taught Reasoner” to leverage visual contrasts to mitigate hallucinations in the model generated rationales. The authors construct contrastive pairs in different datasets for selected samples which do have a similar enough counterpart within each dataset, respectively. The authors further divide the samples into easy, medium and hard and only keep the medium and hard samples. These samples from different datasets together form a new dataset and the authors leverage this dataset for supervised fine-tuning of VLM. Experimental results show that fine-tuning models in this dataset could improve the performance in hallucination, math and general benchmarks.

**Strengths:**

- The proposed framework works in different VQA tasks, including math reasoning, general tasks, and hallucinations.
- The SFT dataset construction process is intuitive.
- A multi-step prompting strategy for contrasting and rethinking is proposed to properly leverage the “reference information” offered by the contrasted image pairs.

**Weaknesses:**

-	The definition of “self-improving” is not clear to me. The collected public datasets for selecting samples to construct the proposed dataset is quite large (21 datasets). Does evaluating a model fine-tuned in 21 datasets in 5 downstream benchmark datasets really count as self-improving? It’s more a strategy to carefully select external knowledge instead of “self-improving.”

-	If I understand correctly, after the model is tuned on the curated dataset consisting of contrastive pairs, the inference process of the tuned model does not include any contrasting any more. Will the model still perform better after being presented with a pair of images?

-	[line 279-line280, line 910-912] how does the model know which one is correct? Why is it expected to pick a correct one to make the framework work?

**Questions:**

Is it important whether the second contrasted image’s question is correctly answered? How is the correctness of that image? Does the order of the image pairs matter? How many images are ideal for this contrast (e.g., how about contrasting 3 or more images)?

Any statistics on how many images are selected as useful in each of the 21 datasets?

After fine-tuning, will the tuned model exhibit a “contrastive rethinking pattern” even on simple tasks? Will the performance degrade on these simple tasks? Will the model hallucinate a “none-existing contrasted image” during the inference?

During inference, does it require a contrast or not? If yes, what to do when no such pair can be found?

---

> ### Author Response · Authors · 2025-11-21
> **Response for Reviewer EeLq Part 1**
>
> We thank the reviewer for the question regarding problem definition and technical details. We carefully consider each of your concerns and provide detailed explanations below.
>
> >The definition of “self-improving” is not clear to me. The collected public datasets for selecting samples to construct the proposed dataset is quite large (21 datasets). Does evaluating a model fine-tuned in 21 datasets in 5 downstream benchmark datasets really count as self-improving? It’s more a strategy to carefully select external knowledge instead of “self-improving.”
>
> | | MMVP | Hallusioin | MathVista | MathVIsion | MMStar | Avg.|
> |---|---|---|---|---|---|---|
> | Base Model | 70.0 | 53.1 | 68.4 | 24.0 | 61.8 | 55.5|
> | SFT on VisCor55k EXCLUDING Reasoning Part | 74.3 | 54.2 | 65.4 | 19.4 | 59.7| 54.6 |
> | SFT on VisCor55k INCLUDING Reasoning Part | 75.7 | 56.3 | 69.7 | 25.3 | 62.4 | 57.9 |
>
> To directly address the concern, the 21 datasets we utilize are standard, publicly available VQA datasets frequently used for training VLMs, which means that they just introduce marginal knowledge. Our key contribution is not the collection itself, but the methodology how we utilize visual contrast to mitigate visual hallucination and generate high-quality reasoning data. As prompted by Reviewer 4FzS's feedback, we conducted an ablation study where we fine-tuned the model only on the raw Question-Answer pairs from these 21 datasets, excluding our generated reasoning part. This direct fine-tuning yielded no performance improvement. This empirically demonstrates that the significant gains from our method do not come from simply exposing the model to more "external knowledge." Instead, the crucial element is the high-quality reasoning paths produced by our self-improving process.
>
> Our definition of "self-improving" is directly inherited from the work of STaR (Self-Taught Reasoner). The STaR defines a self-improving framework as a process where a model leverages its own abilities to generate better reasoning paths on existing data, which in turn are used to improve the model itself. The key idea, as stated by the authors of STaR, is to let a model "improve itself by learning from its own generated reasoning" (STaR, Abstract) and "use its language modeling capacity to improve itself" (STaR, Sec. 1). Crucially, in the STaR framework, the model starts with a large dataset of problems without rationales. It then uses its own innate capability to generate rationales for this dataset. Our work exactly follow the same setting. The 21 datasets we use serve the same role as the large dataset without rationales in the original STaR paper: they provide a diverse environment of problems. The "self-improving" aspect comes from how we generate improved reasoning data within this environment.
>
> In STaR, the intrinsic property leveraged is the model's ability to generate a plausible rationale when given a hint (the correct answer). This is what they term "rationalization" (STaR, Sec. 3.2), which allows the model to generate a "useful rationale" for problems it initially failed. In our VC-STaR, the intrinsic property we leverage is the model's ability to perceive fine-grained visual differences when presented with contrastive VQA pairs. This forces the model to critically re-examine the visual evidence and identify the specific features that differentiate a correct assertion from a hallucinated one. This 'think-contrast-rethink' process is our novel mechanism for enabling the model to generate its own high-quality, hallucination-free reasoning paths.
>
> >If I understand correctly, after the model is tuned on the curated dataset consisting of contrastive pairs, the inference process of the tuned model does not include any contrasting any more. Will the model still perform better after being presented with a pair of images?
>
> Yes, you are absolutely correct. The inference process of our tuned model is standard and does not involve any explicit contrastive steps. As for the question: will the model still perform better after being presented with a pair of images? We have conducted a toy validation on MMVP (pair accuracy), the base model's performance are 48 (Naive) and 58(Contrast). After training, the model performance are 56(Naive) and 70.67(Contrast). This demonstrates that the model after training can still perform better with a pair of images (70.67 v.s 58). The setting is the same as the setting in Fig 1.

---

> > ### Author Response · Authors · 2025-11-21
> > **Response for Reviewer EeLq Part 2**
> >
> > >[line 279-line280, line 910-912] how does the model know which one is correct? Why is it expected to pick a correct one to make the framework work?
> >
> > Sorry for the ambiguity. We would like to clarify the detailed implementation. Here, the 'correct answer' means the ground truth answer, instead of the answer that determined correct by the model. The 'correct answer' serves as a hint to enable VLMs give a better reasoning path with asnwer. In lines 910-912, the actual task for the model is to analyze the consistency of the given ground truth answers between the two VQA in individual pairs and ground this analysis in visual evidence. If the answers are the same, we expect the model's contrastive analysis to identify the common visual features that support this shared conclusion. If the answers are different, we expect the analysis to pinpoint the critical visual distinctions that lead to these different outcomes.
> >
> > In detailed implementation, the placeholders (\<Answer\>) in the prompts will be replaced by the ground truth answers of the paired samples. As the raw samples are sampled from the drawn from established, widely-used VQA datasets, the ground truth answers are easy to obtain.
> >
> > >Is it important whether the second contrasted image’s question is correctly answered? How is the correctness of that image? Does the order of the image pairs matter? How many images are ideal for this contrast (e.g., how about contrasting 3 or more images)?
> >
> > First, regarding the correctness of the contrasted image's QA pair: as we clarified in our previous response, both VQA samples in a pair are sourced from widely-used, established datasets. We therefore operate under the assumption that their ground-truth answers are reliable. However, the reviewer's underlying question is what happens if the quality of the contrastive VQA pair is poor. In fact, our analysis presented in Figure 6 already provides an empirical answer to this question. When we conducted VC-STaR on the DOCCI dataset, the Question-Answer pairs for the contrastive samples were automatically generated from dense captions. In this setting, the quality and correctness of the QA pairs cannot be guaranteed to the same extent as in human-annotated datasets. As discussed in our original analysis of Figure 6, the results on DOCCI showed more limited improvement precisely because the lower quality of these machine-generated QA pairs introduced noise into the contrastive learning process. This empirically validates that high-quality counterparts are indeed important.
> >
> > Second, concerning the order of the image pairs: the order is not a significant factor in our framework. During our pair construction process, every VQA sample has the opportunity to be used to generate one reasoning sample with the help of another counterpart. A given VQA sample can act as an 'anchor' being assisted by a counterpart in one instance, and later serve as a 'counterpart' itself to assist another anchor in generating its reasoning path. This symmetrical role within the overall dataset creation scheme means that a fixed, rigid ordering is not a constraint.
> >
> > | Base Model | 1 Counterpart | 2 Counterpart |
> > |---|---|---|
> > | 45.4 | 53.7(+8.3) | 54.7(+9.3) |
> >
> > Finally, on the ideal number of images for contrast, we have conducted experiments on the GQA dataset, comparing the performance of using a single contrastive counterpart versus using two contrastive counterparts. Considering that the substantial additional computational cost of searching for multiple counterparts did not yield significant performance gains (+8.3 v.s +9.3), we adopted the more efficient and scalable strategy of using a single counterpart for our final data construction.
> >
> > >Any statistics on how many images are selected as useful in each of the 21 datasets?
> >
> > The line graph in Fig.7 illustrates the proportion of data from each dataset that we retained to enter our contrastive pipeline.

---

> > > ### Author Response · Authors · 2025-11-21
> > > **Response for Reviewer EeLq Part 3**
> > >
> > > >After fine-tuning, will the tuned model exhibit a “contrastive rethinking pattern” even on simple tasks? Will the performance degrade on these simple tasks? Will the model hallucinate a “none-existing contrasted image” during the inference?
> > >
> > > The "think-contrast-rethink" procedure is a scaffolding used to create high-quality training data. The "rethink" step is explicitly designed to integrate the insights from the contrastive analysis with the original reasoning into a single, coherent, and self-contained reasoning path for the original image. A key part of this process is ensuring that the final, 'rethought' rationale does not contain any reference to the contrasted image or the contrastive process itself. Consequently, the model is fine-tuned exclusively on these clean, standard-format rationales. The final trained model is, therefore, a conventional VLM with enhanced reasoning capabilities. During inference, it operates in a standard manner: it takes a single image and a question as input and directly generates a reasoning path and an answer. It does not require any contrastive input, hence it hardly attempts to perform a "rethinking" step.
> > >
> > > ---
> > > We sincerely apologize, but due to time constraints, we have not yet revised the PDF document. We will update the manuscript as soon as possible and will leave a comment to guide the reviewer to the changes once it is uploaded.

---

### Official Review · Reviewer_RMKH · 2025-11-01

**Soundness:** 2
**Presentation:** 3
**Contribution:** 3
**Rating:** 6
**Confidence:** 4

**Summary:**

Authors propose Visual Contrastive Self-Taught Reasoner (VC-STaR), a novel self-improving framework that leverages visual contrast to mitigate hallucinations in model-generated rationales.

**Strengths:**

1. The ideas presented are interesting, and the proposed approach has potential to spark further research.

2. The article is well-structured and easy to read, providing a smooth reading experience.

**Weaknesses:**

1. Unclear Motivation for Using Input Comparison to Solve the Hallucination Problem: While the paper proposes using input comparison to mitigate hallucination issues, the motivation or theoretical basis for this approach is not clearly explained. Could the authors elaborate on why this method is effective from a cognitive perspective? For instance, are there relevant studies from cognitive models or psychological theories that can support the effectiveness of this approach?

2. The Problem Being Addressed is Not Clearly Defined: I understand that the goal is to use the reasoning dataset generated after rethinking to train the model. However, my question is: compared to previous reasoning datasets, what exactly is the advantage of your approach? Are there hallucinations in the reasoning data from existing datasets, or is it because you are using a contrastive dataset that allows you to fine-tune the data in a new way to improve the model’s generalization? If it’s the former, are you addressing the problem that existing reasoning datasets are inadequate? If it’s the latter, you need to explain this in more detail and position it as a new contribution.

3. The Inference Method is Not Clearly Defined: How is the trained model used after training? Does it follow the same inference paradigm as before, or does it adopt the contrastive reasoning method shown in Figure 1b? If it’s the latter (option b), does the model need to search for relevant question-answer pairs during inference? Additionally, since you modify the user’s query during rethinking, how does the model generate an appropriate response from the modified query? If it’s not option b, then your inference method hasn’t changed and you are simply training a better model with higher-quality data, correct?

4. Lack of Experimental Details and Limited Experiments: The experimental section lacks sufficient details, and the number of experiments is limited. The specific VLMs used are not mentioned, and the experiments could benefit from being conducted on a wider range of benchmark datasets. Providing more details on experimental settings and parameters would enhance the credibility and reproducibility of the results.

5. Variability in Benchmark Experiment Results: In some benchmarks, the results show significant improvements, while in others, the improvements are small. Has there been any analysis of the reasons behind these discrepancies? It would be helpful if the authors could provide an explanation or discussion regarding the variation in results across different benchmarks.

**Questions:**

see the weaknesses

---

> ### Author Response · Authors · 2025-11-21
> **Response for Reviewer RMKH Part 1**
>
> We thank the reviewer for the insightful questions and appreciate their recognition of our work's potential to spark further research. We try our best to address each weakness:
>
> >Unclear Motivation for Using Input Comparison to Solve the Hallucination Problem: While the paper proposes using input comparison to mitigate hallucination issues, the motivation or theoretical basis for this approach is not clearly explained. Could the authors elaborate on why this method is effective from a cognitive perspective? For instance, are there relevant studies from cognitive models or psychological theories that can support the effectiveness of this approach?
>
> We try to explain the motivation of our work from the cognitive perspective: Learning and reasoning are inherently comparative and contrastive processes. Humans rarely learn concepts in isolation. Instead, we refine our understanding by comparing examples, identifying distinguishing features, and reasoning through analogies and differences. The prototype theory concludes this cognitive behavior as that our human-beings identify new identities by comparing them with the prototype concept [1]. Bedisdes, the structure-mapping theory says that analogical reasoning can recognize the relationships shared by two domains [2]. This mapping can be treated as a fine-grained contrasting process. In our work, the contrastive VQA pair can provide the prototype of the visual concept and reinforce the structur-mapping by generating new reasoning paths via contrasting.
>
> [1] Rosch, Eleanor. "Cognitive representations of semantic categories." Journal of experimental psychology: General 104.3 (1975): 192.
>
> [2] Gentner, Dedre. "Structure-mapping: A theoretical framework for analogy." Cognitive science 7.2 (1983): 155-170.
>
> >The Problem Being Addressed is Not Clearly Defined: I understand that the goal is to use the reasoning dataset generated after rethinking to train the model. However, my question is: compared to previous reasoning datasets, what exactly is the advantage of your approach? Are there hallucinations in the reasoning data from existing datasets, or is it because you are using a contrastive dataset that allows you to fine-tune the data in a new way to improve the model’s generalization? If it’s the former, are you addressing the problem that existing reasoning datasets are inadequate? If it’s the latter, you need to explain this in more detail and position it as a new contribution.
>
> We thank that the reviewer allows us to clarify the precise nature of our contribution. To be direct, our primary contribution is NOT a new contrastive training framework. As we state in line 305 of our manuscript, after generating the reasoning data, we follow a standard fine-tuning procedure. Our novelty lies entirely in the self-improving methodology for generating the reasoning dataset itself by visual contrast. The key advantage of our approach, compared to many existing reasoning datasets, is that it is a self-improving paradigm rather than one based on distillation from a stronger teacher model. We believe this self-evolutionary approach offers a more sustainable path toward building progressively more capable models.
>
> To elaborate on this distinction, many state-of-the-art reasoning datasets (e.g., LLaVA-CoT from GPT-4o, R1-Onevision and LPT from DeepSeek-R1) are constructed by distilling rationales from more powerful, often proprietary models. While effective, this paradigm is fundamentally about imitation, teaching a student model to replicate the reasoning patterns of an existing teacher. The student's performance is therefore inherently capped by the teacher's capabilities and biases. In contrast, our VC-STaR framework enables a model to achieve genuine self-breakthroughs without relying on a superior external teacher. By forcing the model to critically evaluate its own reasoning paths through visual contrast, it learns a deeper, more robust understanding via visual reasoning. Furthermore, when compared to other self-improving approaches such as the original STaR, they often show strong performance on specific tasks but struggle to achieve consistent gains across diverse benchmarks. As shown in Table 1, our contrastive method is the only one in our comparison that demonstrates versatile and consistent positive improvements across all tested categories (hallucination, math, and general reasoning).
>
> In summary, our key contribution is a self-improving methodology that uses visual contrast as a mechanism for self-correction and improvement. This offers a more scalable and sustainable path toward building progressively more capable models.

---

> > ### Author Response · Authors · 2025-11-21
> > **Response for Reviewer RMKH Part 2**
> >
> > >The Inference Method is Not Clearly Defined: How is the trained model used after training? Does it follow the same inference paradigm as before, or does it adopt the contrastive reasoning method shown in Figure 1b? If it’s the latter (option b), does the model need to search for relevant question-answer pairs during inference? Additionally, since you modify the user’s query during rethinking, how does the model generate an appropriate response from the modified query? If it’s not option b, then your inference method hasn’t changed and you are simply training a better model with higher-quality data, correct?
> >
> > We thank the reviewer for the opportunity to clarify our inference method. Our trained model, VC-STaR, actually follows the standard inference paradigm of VLMs, and the contrastive reasoning pipeline shown in Figure 1b is exclusively a part of our offline data generation process. Therefore, during inference, VC-STaR simply takes a user-provided image with a question as input and generates a thinking path then the answer without the need of searching pairs. We will modify the presentation to explicitly define our inference method.
> >
> > Besides, we would like to clarify that our primary contribution is more than simply training a better model with higher-quality data. While the concept of training a model on high-quality data may sound simple, the core challenge is how to obtain such high-quality reasoning data at scale without extensive human annotation. This is an exceptionally difficult problem. Many prior self-improving methods are all attempts to solve this exact challenge. Our work, VC-STaR, proposes a novel and effective solution to this very problem by leveraging a model's intrinsic visual contrastive ability to self-correct its own reasoning flaws.
> >
> > >Lack of Experimental Details and Limited Experiments: The experimental section lacks sufficient details, and the number of experiments is limited. The specific VLMs used are not mentioned, and the experiments could benefit from being conducted on a wider range of benchmark datasets. Providing more details on experimental settings and parameters would enhance the credibility and reproducibility of the results.
> >
> > Regarding the specific VLMs used, we would like to clarify that we stated in line 321 of our manuscript that our main experiments are based on the self-improvement of Qwen2.5VL-7B. Furthermore, as shown in Table 2, we have already validated our method's effectiveness on a range of models, including the smaller Qwen2.5VL-3B and the architecturally different InternVL2.5-8B. The consistent improvements observed across these diverse VLMs demonstrate the generalizability and robustness of our approach. We will try to make this information more clear.
> >
> > We also appreciate the reviewer's suggestion to include a wider range of benchmarks. Upon reviewing our evaluation, we noticed that while our other categories are supported by two benchmarks, the 'General' category in Table 1 currently has only one. To address this and strengthen our evaluation, we have conducted new experiments on a recently proposed general-purpose benchmark, MME-Realworld. Due to the limited rebuttal period, we have initially tested the performance of VC-STaR (59.3), where it achieves a +3.4 point improvement over the base model (55.9). We are committed to testing the other baselines on MME-Realworld and will update Table 1 accordingly for the final version of the paper. This will ensure that all three categories in our main results table are robustly supported by two distinct benchmarks.
> >
> > >Variability in Benchmark Experiment Results: In some benchmarks, the results show significant improvements, while in others, the improvements are small. Has there been any analysis of the reasons behind these discrepancies? It would be helpful if the authors could provide an explanation or discussion regarding the variation in results across different benchmarks.
> >
> > Enhancing model's reasoning capabilities often creates distinct performance biases across different sub-abilities. This is analogous to the trend seen with the latest VLMs (e.g., like https://github.com/QwenLM/Qwen3-VL), "think" versions optimized for complex reasoning exhibit different strengths and weaknesses compared to their instruct version. The core of our work is designed to enhance visual reasoning, which can be evaluated by the hallucination benchmarks and math benchmarks. We include the MMStar benchmark into our evaluation to show that while VC-STaR substantially boosts perception and visual reasoning, it does so without degrading performance on general-purpose tasks. An example in the Table 1 of our paper is R1-Onevision, which excels on the math-centric MathVision benchmark but suffers a catastrophic performance drop on MMStar. Our method successfully avoids this pitfall.

---

> > > ### Author Response · Authors · 2025-11-21
> > >
> > > We sincerely apologize, but due to time constraints, we have not yet revised the PDF document. We will update the manuscript as soon as possible and will leave a comment to guide the reviewer to the changes once it is uploaded.

---

> ### Comment · Reviewer_RMKH · 2025-11-26
>
> Thank you for your detailed rebuttal and clarifications. I appreciate the authors’ efforts in addressing the comments. I do not have further questions. I will maintain my score.

---

### Official Review · Reviewer_EV9r · 2025-11-01

**Soundness:** 3
**Presentation:** 3
**Contribution:** 3
**Rating:** 6
**Confidence:** 5

**Summary:**

This paper addresses the challenge of improving visual reasoning in vision-language models, where conventional self-improving methods fail to correct visual hallucinations in reasoning paths. To overcome this limitation, the authors propose Visual Contrastive Self-Taught Reasoner (VC-STaR), a novel framework that leverages contrastive visual question-answering pairs to refine model-generated rationales. VC-STaR constructs a new dataset, VisCoR-55K, by curating contrastive VQA pairs across multiple domains and generating faithful visual rationales through a “think–contrast–rethink” pipeline. Fine-tuning VLMs on VisCoR-55K significantly enhances their reasoning performance across five benchmarks, surpassing existing self-improving baselines and models trained on state-of-the-art reasoning datasets. These results demonstrate that contrastive learning enables VLMs to self-correct visual hallucinations and advance visual reasoning.

**Strengths:**

1)	This paper proposes a task-agnostic, three-stage pipeline to construct contrastive pairs, with principled text/vision similarity thresholds to ensure both semantic anchoring and visual proximity.
2)	The paper takes an insightful contrastive perspective, showing that learning from visual contrasts effectively reduces hallucinations and strengthens reasoning accuracy in VLMs.
3)	Analyses isolate the impact of curation strategies, sample difficulty, and pair types, producing actionable design guidance.

**Weaknesses:**

1)	In Table 1, the method fails to achieve top results on MMStar. While it improves over its backbone, it still lags behind several baselines, and the paper does not explain why its gains do not generalize to this benchmark.
2)	In line 258, the authors state that only median-difficulty contrastive VQA pairs are retained for rationale generation. However, the paper does not discuss why hard samples are excluded or how they might be addressed. Exploring strategies for handling these challenging cases could further enhance the model’s robustness and generalization.
3)	In Figure 1b, the meaning of Setting C is not clearly explained. Although readers can infer its general idea, the paper should explicitly define it to avoid ambiguity and ensure the experimental setup is fully transparent.
4)	The paper’s main contribution lies in constructing the VisCoR-55K dataset, yet neither the dataset nor the accompanying code is released. Without open access, it is difficult to verify the results or reproduce the data-curation process, which weakens the paper’s practical impact and credibility.
5)	The additional data generation and contrastive fine-tuning steps likely increase training cost substantially, but the paper does not report resource usage or efficiency.

**Questions:**

Please refer to the weak points.

---

> ### Author Response · Authors · 2025-11-21
> **Response for Reviewer EV9r Part 1**
>
> We sincerely thank the reviewer for their thorough and constructive feedback. We are encouraged that the reviewer found our perspective is insightful. We appreciate the detailed suggestions for improvement, which we believe will significantly strengthen our paper. Below, we address each of the weaknesses point-by-point.
>
> > In Table 1, the method fails to achieve top results on MMStar. While it improves over its backbone, it still lags behind several baselines, and the paper does not explain why its gains do not generalize to this benchmark.
>
> VC-STaR improves over its base model, but it does not achieve the top score among all baselines. This phenomenon is not a weakness of our method. The core of our work is designed to enhance visual reasoning, which is not the major consideration of the MMStar benchmark. We include the MMStar benchmark into our evaluation to show that while VC-STaR substantially boosts perception and visual reasoning, it does so without degrading performance on general-purpose tasks. A prominent example in the Table 1 of our paper is R1-Onevision, which excels on the math-centric MathVision benchmark but suffers a catastrophic performance drop on MMStar. Our method successfully avoids this pitfall.
>
> We are grateful for the reviewer's suggestion, and we will revise the discussion  of our paper to include the detailed analysis above, making our design philosophy and the context of the MMStar results explicit. At the same time, we would like to respectfully reiterate that the central goal of our work is to enhance perception and reasoning capability. The inclusion of the MMStar benchmark serves primarily to demonstrate that our method achieves this specialized improvement while preserving general capabilities, thus offering a more balanced and versatile model.
>
> > In line 258, the authors state that only median-difficulty contrastive VQA pairs are retained for rationale generation. However, the paper does not discuss why hard samples are excluded or how they might be addressed. Exploring strategies for handling these challenging cases could further enhance the model’s robustness and generalization.
>
> Thanks for your valuable seggestion. The exclusion of "hard" samples is a principled decision central to our goal: generating reliable reasoning paths for robust self-improvement. As defined in the paper (Lines 257-258), "hard" samples are those for which the model still fails to produce the correct answer even after undergoing our full 'think-contrast-rethink' pipeline. Incorrect final answers signify unreliable or flawed reasoning paths. Incorporating such data into our self-improving loop would introduce significant noise, ultimately result in the unrobust learning process. Therefore, it is a natural process to exclude the hard samples in our curation pipeline.
>
> This data filtering strategy is, in fact, a widely adopted practice in self-improving frameworks to ensure stable and effective learning. For instance, the STaR employs a similar principle: it explicitly discards hard samples with incorrect answers even prompted with hints, training only on the successful ones. Our strategy is conceptually a similar version of this approach, prioritizing the quality of the training signal.
>
> Nevertheless, we fully agree with the reviewer that exploring strategies for handling these challenging hard cases is a highly constructive and important direction. These hard samples represent the current limitations of the model's reasoning ability, and effectively leveraging them would be crucial for pushing the boundaries of its robustness and generalization. We believe that future research into more advanced techniques for correcting or learning from these difficult examples will further enhance the model’s robustness and generalization.
>
> >In Figure 1b, the meaning of Setting C is not clearly explained. Although readers can infer its general idea, the paper should explicitly define it to avoid ambiguity and ensure the experimental setup is fully transparent.
>
> Setting C refers to providing a given VQA sample with its contrastive counterpart for answering the question. This is just the setting of answering process at the right of Fig1(a) in our paper. We appreciate the reviewer highlighting this. In our revised manuscript, we will update the information of Setting C to make it more clear.

---

> > ### Author Response · Authors · 2025-11-21
> > **Response for Reviewer EV9r Part 2**
> >
> > >The paper’s main contribution lies in constructing the VisCoR-55K dataset, yet neither the dataset nor the accompanying code is released. Without open access, it is difficult to verify the results or reproduce the data-curation process, which weakens the paper’s practical impact and credibility.
> >
> > We thank the reviewer for emphasizing the importance of reproducibility. As stated in line 29 of our submitted manuscript, we have already made a promise to publicly release the VisCoR-55K dataset upon the publication of our paper. Our institutional policy does not permit the release of data until the paper has been accepted. Therefore, we are committed to making the data publicly available immediately upon the paper's acceptance.
> >
> > >The additional data generation and contrastive fine-tuning steps likely increase training cost substantially, but the paper does not report resource usage or efficiency.
> >
> > We thank the reviewer for this insightful comment and agree that reporting resource usage is crucial. Our approach introduces a modest cost in data generation, while maintaining a fine-tuning cost comparable to existing self-improving frameworks like STaR.
> > - Fine-tuning Cost: We conduct standard Supervised Fine-Tuning (SFT) on the 7B model using the LLaMAFactory framework once the VisCoR-55K dataset is generated. This process takes approximately four hours on a single node with 8x A800 GPUs. This cost is nearly identical to the standard fine-tuning procedure in approach like STaR.
> > - Data Generation Cost: The additional cost stems from our data generation pipeline. The one-time contrastive VQA pair mining step (L238-L251) processes approximately 7 samples per second；The 'rethink' step (L281-L290) generates reasoning paths at about 1 sample per second.
> > We argue this is a reasonable and acceptable investment to create a large-scale, high-quality reasoning dataset. To relieve this generation cost for the community and foster future research, we have committed to releasing the full VisCoR-55K dataset, including the intermediate contrastive VQA pairs.
> >
> > Furthermore, we have already implemented optimizations to accelerate the generation, including multi-processing concurrency, batch-wise computation for pair mining in a large-scale raw dataset, and leveraging the vLLM framework for fast inference. While our current implementation is effective, we believe there is still room for further engineering optimizations to improve efficiency for further scaling up. We will add the discussion about the resource usage into our paper.
> >
> > ---
> >
> > We sincerely apologize, but due to time constraints, we have not yet revised the PDF document. We will update the manuscript as soon as possible and will leave a comment to guide the reviewer to the changes once it is uploaded.

---

### Author Response · Authors · 2025-12-02
**Summary of the Rebuttal Process**

We thank the reviewers and the ACs for your efforts. To facilitate the following review process, we provide an objective overview of the review and rebuttal process prior to the API security incident.

Our work was ***initially scored 8, 6, 6, 6, 6***, receiving commendations for its novelty and potential to "spark further research." Notably, following our detailed rebuttal, Reviewer 4FzS ***raised the score to 8 (confidence from 4 to 5)*** and Reviewer RMKH expressed satisfaction, stating they ***had no further questions***, both before the 26 Nov.\
\
Our rebuttal addressed key concerns, which we summarize below:

1. ***Method Clarification (RMKH and EeLq):***

    - *Concerns about the position of this work* from RMKH: Our core contribution is to introduce a new self-improving mechanism for visual reasoning, driven by our key finding that VLMs reason more accurately with contrastive inputs. Our VC-STaR puts this insight into practice, allowing models to bootstrap their own reasoning.

    - *Questions about the definition of self-improving* from EeLq: We follow the standard definition of self-improving in previous works, e.g., the STaR. Self-improving framework is a process where a model leverages its own abilities to generate better reasoning paths on existing data, which in turn are used to improve the model itself.

2. ***Inference Paradigm (RMKH and EeLq):***

    - *Concerns about the inference setting* from RMKH and EeLq: We clarified that the contrastive pipeline is exclusively for offline reasoning data generation. This ensures the final model operates with an efficient, standard inference paradigm (without extra contrastive inputs), imposing no runtime overhead from pair searching and ensuring the versatility of our VC-STaR.

3. ***Additional Experiments (RMKH, mDXQ, and 4FzS):***

    - *Request for experiments on more benchmarks* from RMKH and mDXQ: We report new results on MME-RealWorld, a recent and challenging benchmark, to provide a more comprehensive assessment of our method's generalization.

    - *Curiosity about a baseline that trains models solely with instruction–response pairs* from 4FzS: We include this baseline to demonstrate that the effectiveness of our VC-STaR stems from the generated rationales rather than the instruction-response pairs.

4. ***Computational Cost (EV9r and 4FzS) and Reproducibility (EV9r):***

    - *Questions about extra computational burden relative to previous methods* from EV9r and 4FzS: Our approach introduces a modest cost in data generation, while keeping a comparable fine-tuning cost to existing self-improving frameworks like STaR. The revised manuscript now includes a detailed analysis of the computational overhead.

    - *Concern about the reproducibility* from EV9r: To ensure full reproducibility and foster future research, we have committed to the public release of our VisCoR-55K dataset and all source code upon acceptance.

5. ***Others***

    Regarding the detailed concerns about *our experimental setup* (EV9r and mDXQ), the *discussion from a cognitive perspective* (RMKH), and the *embedding model choice* (mDXQ), we have both provided point-by-point responses and made corresponding revisions in the updated manuscript.
\
We have carefully considered all comments and revised our submission accordingly. We are confident that these updates further solidify the contribution of our work.

We sincerely thank the reviewers, ACs, and PCs again for your engagement, contributions, and valuable feedback.

---

### Meta-Review · Area_Chair_DCkZ · 2026-01-04

**Summary:**

The main strengths of this paper are (1) the key idea of leveraging contrastive visual pairs to improve reasoning fidelity is intuitive and well-motivated, (2) the proposed contrast-based self-improving framework is task-agnostic and principled, (3) the data curation is scalable and principled, leading to a high-quality visual reasoning dataset, (4) fine-tuning on VisCoR-55K yields significant performance gains across hallucination, math, and general benchmarks. The major weaknesses or concerns are (1) the definition of self-improving and the key contribution are not clear, where the authors clarified that the definition follows the standard of self-improving in previous works, and the key contribution is to introduce a new self-improving mechanism for visual reasoning motivated by the key findings, (2) the inference method is not clearly defined, where the authors clarified that the contrastive pipeline is for offline reasoning data generation without extra runtime overhead, (3) missing experiments, computational cost, and reproducibility, which are addressed with further results and the commit of public release.

This paper received 8, 6, 6, 6, 6 as initial scores. Reviewer 4FzS further raised the score from 6 to 8 and confidence from 4 to 5 after discussion. The area chair thinks the main concerns have been well addressed, and the insights of contrast-based reasoning and contributions of dataset and approach are significant. The final decision is accept.

**Reviewer Concerns:**

The major weaknesses or concerns are (1) the definition of self-improving and the key contribution are not clear, where the authors clarified that the definition follows the standard of self-improving in previous works, and the key contribution is to introduce a new self-improving mechanism for visual reasoning motivated by the key findings, (2) the inference method is not clearly defined, where the authors clarified that the contrastive pipeline is for offline reasoning data generation without extra runtime overhead, (3) missing experiments, computational cost, and reproducibility, which are addressed with further results and the commit of public release.

**Reviewer Scores:**

Reviewer 4FzS further raised the score from 6 to 8 and confidence from 4 to 5 after discussion. Reviewer RMKH had no further questions and maintained the score. Since the concerns were well addressed, the final scores would be all positive.

---

### Decision · Program_Chairs · 2026-01-26

Accept (Oral)